# MaxSup: Fixing Label Smoothing for Improved Feature Representation

## Abstract

Label Smoothing (LS) aims to prevent Neural Networks from making overconfident predictions and improve generalization. Due to its effectiveness, it has become an indispensable ingredient in the training recipe for tasks such as Image Recognition and Neural Machine Translation. Despite this, previous work shows it encourages an overly tight cluster in the feature space, which 'erases' the similarity information of individual examples. A more recent study empirically shows that LS also makes the network more confident in its wrong predictions. By isolating the loss induced by Label Smoothing into a combination of a regularization term and an error-enhancement term, we reveal the underlying mechanism behind such defects of Label Smoothing. To remedy this, we present a solution called Max Suppression (MaxSup), which consistently applies the intended regularization effect during training, independent of the correctness of prediction. By examining the learned features, we demonstrate that MaxSup successfully enlarges intra-class variations, while improving inter-class separability. We further conduct experiments on Image Classification and Machine Translation tasks, validating the superiority of Max Suppression. The code implementation is available at `https://anonymous.4open.science/r/Maximum-Suppression-Regularization-DB0C`.

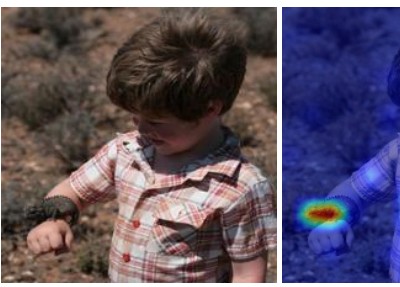 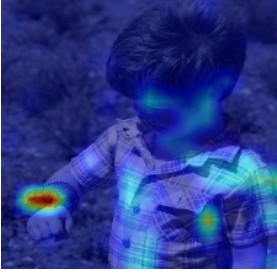 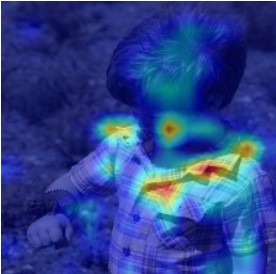

| African chameleon | CAM with MaxSup | CAM with LS |

## 1 Introduction

In multi-class classification (Russakovsky et al., 2015; LeCun, 1998), different categories are widely represented by one-hot vectors, assuming them to be cardinal and orthogonal. However, many classes often share common low-level features (Zeiler and Fergus, 2014; Silla and Freitas, 2011) or high-level similarities (Chen et al., 2021; Yi et al., 2022; Novack et al., 2023). The assumption of orthogonality underlying the one-hot labels apparently deviates from this observation, which tends to produce over-confident classifiers with reduced generalization ability (Guo et al., 2020).

To prevent the network from being over-confident about its predictions and thus generalize better, Szegedy et al. (2016) proposed Label Smoothing (LS), which replaces the one-hot label with a convex combination of the original label and a vector of ones. Thanks to its simplicity and effectiveness, it has been widely adopted for tasks such as Image Recognition (He et al., 2016; Touvron et al., 2021; Liu et al., 2021; Zhou et al., 2022b) and Neural Machine Translation (Gao et al., 2020; Alves et al., 2023). Despite the improved classification performance, Müller et al. (2019) identified an inherent

flaw in Label Smoothing: it tends to compress samples of the same class into **overly tight clusters in the feature space**, which consequently 'erases' the similarity information that an individual example has to different classes. Such information loss might not be well reflected in the classification performance, but it potentially **harms the effectiveness of the learned representation in broader downstream applications**, such as linear transfer accuracy (Kornblith et al., 2021). More recently, Zhu et al. (2022) empirically identified that **Label Smoothing results in more confident errors**, but the reason behind such an issue is not yet understood.

In this paper, we reveal that the part of the training objective introduced by Label Smoothing surprisingly contains two problematic components: a regularization component that only functions as expected when the predictions are correct, and an error-enhancement term that emerges when the predictions are incorrect, encouraging the network to become overconfident in its wrong predictions. In line with Zhu et al. (2022), the term "overconfidence" in our work specifically refers to the network's confidence in its top-1 prediction, which is different from the overconfidence in the context of model calibration. This work **uncovers the underlying mechanism of the recently observed defect of Label Smoothing (Zhu et al., 2022), and shows that it is also the cause of the overly tight clusters**.

In light of this observation, we further propose a solution called Max Suppression (MaxSup), which consistently applies the intended regularization effect during training, regardless of whether the prediction is correct or not. The quantitative evaluation of the features from the penultimate layer highlights that MaxSup successfully allows for a larger intra-class variation, while improving the inter-class separability in the feature space compared to Label Smoothing. The improved performance on Image Classification and Machine Translation tasks additionally supports that Max Suppression is a superior alternative to Label Smoothing.

Our contributions are as follows:

- We reveal the underlying mechanism of the previously observed defects of Label Smoothing, highlighted by the Inconsistent Regularization term as well as the Error-Enhancement term via our novel decomposition of the training objective.
- We propose Max Suppression as a closed-form solution to the identified issue, which is demonstrated to be a superior alternative to Label Smoothing.
- We show that training with Max Suppression not only improves the classification performance, but also better retains the similarity information of individual samples to different classes.

## 2 RELATED WORK

### 2.1 REGULARIZATION

Regularization techniques aim to enhance the generalization ability of deep neural networks. L2 (Krogh and Hertz, 1991) and L1 (Zou and Hastie, 2005) Regularization control model complexity by penalizing large or sparse weights, respectively. Dropout (Srivastava et al., 2014) randomly deactivates neurons during training, helping to reduce over-fitting by preventing co-adaptation of features. Loss-based regularization techniques, such as Label Smoothing (Szegedy et al., 2016), soften target labels to mitigate overconfidence in predictions, which leads to more accurate and better calibrated classifiers (Müller et al., 2019). To exploit the clues in the model's prediction, Zhang et al. (2021); Liang et al. (2022) further introduced Online Label Smoothing (OLS) and Zipf Label Smoothing (Zipf-LS) to replace the uniform distribution with the predicted distribution based on the previous and current model weights, respectively. Other approaches, like Confidence Penalty (Pereyra et al., 2017), directly penalize overly confident outputs to enhance model calibration. Moreover, Logit penalty (Dauphin and Cubuk, 2021) that minimizes the l2-norm of the logits is also shown to be effective (Kornblith et al., 2021).

### 2.2 STUDY ON LABEL SMOOTHING

A line of studies investigates Label Smoothing in the context of knowledge distillation: Yuan et al. (2020) revealed the underlying connection between Label Smoothing and Knowledge Distillation,

Shen et al. (2021) provided a comprehensive evaluation of the compatibility between Label Smoothing and Knowledge Distillation, and Chandrasegaran et al. (2022) emphasized the importance of using an LS-trained teacher with a low-temperature transfer. Kornblith et al. (2021) empirically validated that Label Smoothing leads to increased tightness and separation of feature clusters, as well as degraded transfer learning performance. The impact of Label Smoothing on the learned feature space is also investigated in the context of neural collapse (Zhou et al., 2022a; Guo et al., 2024), by examining the properties of feature clusters.

# 3 MAX SUPPRESSION REGULARIZATION

In this section, we begin by disentangling the training objective into two components: the standard Cross-Entropy loss with one-hot labels and Label Smoothing (LS) loss. We then focus on the LS loss component, reformulating it at the logit level for a clearer understanding of its internal mechanisms. This logit-level formulation allows us to further decompose LS into two key terms: a Regularization term and an Error-Enhancement term. Based on this decomposition, we highlight the limitations of LS, particularly its tendency to amplify errors through the Error-Enhancement term. To address these limitations, we propose *Max Suppression Regularization* (MaxSup) as a remedy.

## 3.1 REVISITING LABEL SMOOTHING

Label Smoothing (LS) is a commonly used regularization technique to prevent models from becoming overly confident in their predictions. Instead of assigning a probability of 1 to the ground-truth class and 0 to all other classes, LS smooths the target distribution by distributing a small portion of the probability mass uniformly across all classes. Below is the formal definition:

**Definition 3.1.** For a classification task with $K$ distinct classes, Label Smoothing transforms a one-hot encoded label $\mathbf{y} \in \mathbb{R}^K$ into a soft label $\mathbf{s} \in \mathbb{R}^K$ by taking a convex combination of $\mathbf{y}$ and a uniform distribution over all classes:

$$s_k = (1 - \alpha)y_k + \frac{\alpha}{K} \tag{1}$$

where $y_k = \mathbb{1}_{k=gt}$, i.e., $y_k = 1$ if class $k$ is the ground-truth class; otherwise $y_k = 0$. The scalar $\alpha \in [0, 1]$ is the smoothing weight, and $gt$ denotes the index of the ground-truth class. Label Smoothing assigns a portion of the probability mass $\frac{\alpha}{K}$ uniformly across all non-ground-truth classes while reducing the probability of the ground-truth class by a factor of $\alpha$.

To analyze the effect of LS on the training objective, we decompose the Cross-Entropy loss into two parts: the standard Cross-Entropy loss without Label Smoothing and the additional loss term introduced by Label Smoothing:

**Lemma 3.2.** *Decomposition of Cross-Entropy Loss with Soft Label*

$$\underbrace{H(\mathbf{s}, \mathbf{q})}_{\text{CE with Soft Label}} = \underbrace{H(\mathbf{y}, \mathbf{q})}_{\text{CE with Hard Label}} + \underbrace{L_{LS}}_{\text{Label Smoothing Loss}} \tag{2}$$

*where the Label Smoothing Loss $L_{LS}$ is given by:*

$$L_{LS} = \alpha \left( H\left(\frac{\mathbf{1}}{K}, \mathbf{q}\right) - H(\mathbf{y}, \mathbf{q}) \right), \tag{3}$$

*where $\mathbf{q}$ denotes the predicted probability vector, $H(\cdot)$ denotes Cross-Entropy (CE) between two distributions, and $L_{LS}$ indicates the loss component introduced by Label Smoothing, termed Label Smoothing Loss. Note that the original Cross-Entropy Loss $H(\mathbf{y}, \mathbf{q})$ is unweighted by $\alpha$ because the weight is implicitly incorporated in $L_{LS}$. $\frac{1}{K}$ denotes the uniform distribution introduced by Label Smoothing. This decomposition shows that LS not only modifies the ground-truth label but also adds a regularization effect through $L_{LS}$, which encourages a smoother output distribution and helps reduce overfitting.*

*(Please refer to Appendix A for the proof.)*

Based on the decomposition in Lemma 3.2, we further simplify the Label Smoothing Loss into a formulation of logit operations, which allows for a closer inspection of the underlying mechanism

of Label Smoothing. Due to the broad usage of CutMix and Mixup in the training recipe of modern Neural Networks, we additionally take their impact into account together with Label Smoothing. For training a classifier with Label Smoothing, we show that the following holds:

**Theorem 3.3.** *Logit-Level Formulation of Label Smoothing Loss*

1. *Without CutMix or Mixup:*

$$L_{LS} = \alpha \left( z_{gt} - \frac{1}{K} \sum_{k=1}^{K} z_k \right) \tag{4}$$

   *where $L_{LS}$ is the Label Smoothing loss component. This formulation expresses the loss as the difference between the logit corresponding to the ground-truth class $z_{gt}$ and the average of all logits across the $K$ classes, $\frac{1}{K} \sum_{k=1}^{K} z_k$. This shows that LS regularizes the difference between the ground-truth logit and the average logit across all classes, preventing the model from becoming overly confident in its predictions.*

2. *With CutMix and Mixup:*

$$L'_{LS} = \alpha \left( \lambda z_{gt1} + (1 - \lambda) z_{gt2} - \frac{1}{K} \sum_{k=1}^{K} z_k \right) \tag{5}$$

   *where $L'_{LS}$ is the Label Smoothing loss component in the presence of CutMix or Mixup. In this case, $z_{gt1}$ and $z_{gt2}$ are the logits corresponding to the two ground-truth classes introduced by CutMix or Mixup, and $\lambda$ is the mixing ratio between these two classes. The formulation captures how LS smooths the two logits, $z_{gt1}$ and $z_{gt2}$, and applies regularization across all classes.*

*(Please refer to Appendix B for the proof.)*

Depending on whether the logits are larger or smaller than $z_{gt}$, i.e., whether the prediction is correct or not, the Label Smoothing Loss $L_{LS}$ can be further decomposed into two key components: a **Regularization term**, which reduces overconfidence in correct predictions, and an **Error-Enhancement term**, which exacerbates overconfidence in incorrect predictions. Note that the Overconfidence we discuss here is different from the Overconfidence in Model Calibration.

**Corollary 3.4.** *Decomposition of Label Smoothing Loss*

$$L_{LS} = \underbrace{\frac{\alpha}{K} \sum_{z_m < z_{gt}}^{M} (z_{gt} - z_m)}_{Regularization} + \underbrace{\frac{\alpha}{K} \sum_{z_n > z_{gt}}^{N} (z_{gt} - z_n)}_{Error\text{-}Enhancement} \tag{6}$$

*where $M$ and $N$ denote the number of logits smaller than or greater than $z_{gt}$ and $M + N = K - 1$. Note that the second summation term in Equation (6) is always zero except when $z_{gt} \neq z_{max}$, i.e., when the classifier makes a incorrect prediction. (1) **Regularization term** corresponds to the part where logits are smaller than $z_{gt}$ and is always non-negative. (2) **Error-Enhancement term** corresponds to the logits larger than $z_{gt}$ and is non-positive.*

The regularization is intended to prevent a model from overfitting to the training datapoints. For classification problems, this may occur when a model is highly confident on the ground truth training label. However, **overfitting is not occurring when a model is incorrect** as it is inherently not fit to the training data. By penalizing the ground truth logit and enhancing the error on incorrect predictions, label smoothing does not prevent overfitting and instead **worsens the learning on poorly fit or incorrectly classified samples**. Let us consider the following two cases separately:

- **When the network makes a correct prediction**, i.e., $z_{gt} = z_{max}$, the error-enhancement term equals zero, and the regularization term penalizes the network for being over-confident about its prediction (the peak position, i.e., $z_{max}$, is regarded as the prediction of a classifier) as desired.

- **When the network makes an incorrect prediction**, i.e., $z_{gt} \neq z_{max}$, Label Smoothing faces two problems:

1. **Error-Enhancement**: The non-zero error-enhancement term encourages an increase in the gap between the ground-truth logit and the larger logits, further enhancing the over-confidence in the incorrect prediction.

2. **Inconsistent Regularization**: The regularization term $\frac{\alpha}{K} \sum_{z_m < z_{gt}}^{M} (z_{gt} - z_m)$ of LS fails to penalize the network for being over-confident about its prediction (the peak position, i.e., $z_{max}$). Instead, it further **reduces the already underestimated** $z_{gt}$.

Note that concurrent work (Xia et al., 2024) arrives at a similar observation through gradient analysis. The findings from both studies can be seen as mutually validating. However, our decomposition offers an additional advantage, as it allows us to derive MaxSup as a direct solution to the observed problem.

To verify the effects of the different components of Label Smoothing, we conduct an ablation study using the Deit-Small model (Touvron et al., 2021), trained on ImageNet-1K. For clarity and to isolate the impact of Label Smoothing, we remove Mixup and CutMix from the data augmentation pipeline. This allows us to assess the contributions of each component of Label Smoothing in a clean ablation setting. The results are summarized in Table 1.

Table 1: Preliminary study on Label Smoothing Loss components on ImageNet-1K using Deit-Small model as baseline. Note that we remove CutMix&Mixup.

| Method | Formulation | Accuracy |
|---|---|---|
| Baseline | - | 74.21 |
| + Label Smoothing | $\frac{\alpha}{K} \sum_{z_m < z_{gt}}^{M} (z_{gt} - z_m) + \frac{\alpha}{K} \sum_{z_n > z_{gt}}^{N} (z_{gt} - z_n)$ | 75.91 |
| + Regularization | $\frac{\alpha}{M} \sum_{z_m < z_{gt}}^{M} (z_{gt} - z_m)$ | 75.98 |
| + Error-Enhancement | $\frac{\alpha}{N} \sum_{z_n > z_{gt}}^{N} (z_{gt} - z_n)$ | 73.63 |
| + Error-Enhancement | $\alpha(z_{gt} - z_{max})$ | 73.69 |
| + MaxSup | $\alpha(z_{max} - \frac{1}{K} \sum_{k \in K} z_k)$ | 76.12 |

As demonstrated in Table 1, the performance improvements from Label Smoothing are solely attributed to the Regularization term. The Error-Enhancement term, on the other hand, consistently leads to performance degradation. This is evident from the reduced accuracy when only the Error-Enhancement term is applied. For a fair comparison, we use the default smoothing weight $\alpha = 0.1$ from the baseline. The ablation study confirms that the subtraction of the maximum logit ($z_{max}$) is the main cause of the performance drop, as demonstrated by the comparable degradation when only the Error-Enhancement term is included. This indicates that Label Smoothing's effectiveness stems entirely from its Regularization component, while the Error-Enhancement component negatively impacts model performance by increasing overconfidence in incorrect predictions. Moreover, using the regularization term alone (75.98%) only brings marginal improvement (+0.07%) over Label Smoothing (75.91%), whereas MaxSup (76.12%) leads to larger improvement (0.21%) over Label Smoothing (75.91%), supporting our analysis that MaxSup fixes the issues of Label Smoothing by applying the intended regularization and removing the error-enhancement upon incorrect predictions.

## 3.2 MAX SUPPRESSION REGULARIZATION (MAXSUP)

To address the Inconsistent Regularization and Error-Enhancement issue in Label Smoothing, we introduce Max Suppression (MaxSup), which penalizes the max logit instead of the ground-truth logit. By contrasting Equation (6) and Equation (7), it is obvious that MaxSup behaves identically to Label Smoothing when the classifier makes a correct prediction, but crucially, it consistently applies the desired regularization effect and eliminates the Error-Enhancement term for incorrect predictions. In essence, MaxSup penalizes both overfitting on well-fit datapoints (when the prediction is correct with very high confidence), as well as encouraging learning on poorly fit datapoints (when the prediction does not fit the ground truth).

**Definition 3.5. Max Suppression Regularization**

$$L = \alpha(z_{max} - \frac{1}{K} \sum_{k=1}^{K} z_k) \tag{7}$$

For intuitive understanding, we also provide another formulation of the proposed Max Suppression loss by transforming its current logit-level formulation back into the label form in Equation (8). Since a negative amount of the probability mass is assigned to the position with the maximum likelihood, the soft label generated by Max Suppression is no longer a proper distribution. However, it is straightforward to grasp the impact of the negative probability mass, i.e., it consistently prevents the model for being over-confident in its prediction.

**Definition 3.6. Max Suppression Regularization as Label Smoothing**

For the classification of $K$ distinct classes, Max Suppression transforms the one-hot label $\boldsymbol{y} \in \mathbb{R}^K$ into a soft label $\boldsymbol{s} \in \mathbb{R}^K$ via a convex combination of $\boldsymbol{y}$ and a vector with all entries equal to one:

$$s_k = y_k + \frac{\alpha}{K} - \alpha \mathbb{1}_{k=Argmax(\boldsymbol{q})} \tag{8}$$

where $y_k = \mathbb{1}_{k=gt}$ and $\mathbb{1}_{k=gt}$ is an indicator function with the subscript $k$ denoting the $k^{th}$ entry of the label and $gt$ denoting the ground-truth class. Additionally, $\alpha \in [0, 1]$ is the hyperparameter.

We also explore the relationship between Label Smoothing and Max Suppression in terms of their gradients. The analysis shows that Max Suppression Regularization redistributes a gradient of magnitude $\alpha$ between the True Class and the incorrect Prediction Class. Please refer to Appendix C for more details.

## 4 IMPROVED INTRA-CLASS VARIATION AND INTER-CLASS SEPARABILITY

Beyond improving inter-class separability, which enhances classification performance, we argue that the key strength of MaxSup lies in its ability to capture greater intra-class variation—an indicator of improved representation learning. As analyzed in Section 3.1, Label Smoothing only performs the desired regularization on the correct predictions (top-1 probability), whereas MaxSup regularizes both the correct and incorrect predictions (top-1 probability), thereby leaning to even larger inter-class separability. Moreover, MaxSup eliminates the error-enhancement defect of Label Smoothing, which may be the cause of the severely reduced intra-class variance. We validate the improved intra-class variation and inter-class separability using the metrics in Kornblith et al. (2021), and the results are listed in Table 2.

| Methods | $\bar{d}_{\text{within}} \uparrow$ | | $\bar{d}_{\text{total}}$ | | $R^2(1 - \frac{\bar{d}_{\text{within}}}{\bar{d}_{\text{total}}}) \uparrow$ | |
|---|---|---|---|---|---|---|
| | Train | Val | Train | Val | Train | Val |
| Baseline | 0.3114 | 0.3313 | 0.5212 | 0.5949 | 0.4025 | 0.4451 |
| LS | 0.2632 | 0.2543 | 0.4862 | 0.4718 | 0.4690 | 0.4611 |
| MaxSup | 0.2926 (**+0.03**) | 0.2998 (**+0.05**) | 0.6081 (**+0.12**) | 0.5962 (**+0.12**) | 0.5188 (**+0.05**) | 0.4972 (**+0.04**) |
| Logit Penalty | 0.2840 | 0.3144 | 0.7996 | 0.7909 | 0.6448 | 0.6024 |

Table 2: Quantitative measures for inter-class separability and intra-class variation of feature representations, using ResNet-50 trained on ImageNet-1K. Results are provided for Training Set and Validation Set.

The expanded intra-class variation suggests that MaxSup enables the model to capture richer, more detailed similarity information—reflecting how individual examples relate to different classes. In contrast, Label Smoothing tends to 'erase' these finer distinctions, as noted by Müller et al. (2019). It can be further validated by the linear transfer performance (please refer to Table 3) on the CIFAR-10 dataset, using the pretrained ResNet50, following Kornblith et al. (2021). Note that Kornblith et al. (2021) also examines Logit Penalty, a regularizer which is closely related to MaxSup: Logit Penalty regularizes the l2-norm of the logits, whereas MaxSup specifically regularizes the peak logits. Indeed, Logit Penalty imposes much stronger constraints on the logits, since it reduces the absolute magnitudes of individual logits, while MaxSup only encourages the peak logit value to be close to the mean value of all logits. The stronger regularization effect of Logit Penalty leads to larger inter-class separability in Table 2, but performs poorly on both Linear Transfer task in Table 3 and ImageNet Classification task in Table 4.

| Methods | Linear Transfer *val. acc* |
|---|---|
| Baseline | 0.8143 |
| Label Smoothing | 0.7458 |
| Logit Penalty (Dauphin and Cubuk, 2021) | 0.7242 |
| MaxSup | 0.8102 (**+0.06**) |

Table 3: Validation performance of different methods based on multi-nominal Logistic Regression with $l_2$ regularization in CIFAR10 validation set. We searched the strength of the regularization from $1e - 4$ to $1e2$, the search step size is increasing by an order of magnitude.

## 5 EXPERIMENTS

### 5.1 EVALUATION ON IMAGENET CLASSIFICATION

In this section, we evaluate the efficacy of MaxSup, comparing its performance against standard Label Smoothing and its variants on Imagenet-1k.

#### 5.1.1 EXPERIMENT SETUP

**Model Training recipes** We adopt the most representative models for CNNs and Transformers: ResNet families (He et al., 2016), MobileNetV2 (Sandler et al., 2018), and DeiT-Small (Touvron et al., 2021) conducting evaluations on the large-scale ImageNet dataset (Krizhevsky et al., 2012). For ResNet-50 training, we use baseline recipes in TorchVision[1]. Specifically, the ResNet50 model was trained for 90 epochs using stochastic gradient descent (SGD) with a momentum of 0.9 and weight decay of 1e-4. The initial learning rate was set to 0.5, employing a cosine annealing learning rate scheduler. A linear warmup strategy was applied for the first 5 epochs with a warmup decay of 0.01. For regularization, we used a weight decay of 2e-05, while excluding normalization layers from weight decay. For DeiT-Small, we use the official implementation provided by the authors and train the model from scratch without applying knowledge distillation. While knowledge distillation is a prominent feature of the original DeiT paper, we intentionally exclude it in our setup to ensure a clear and unbiased reflection of MaxSup's performance.

**Hyperparameters for Methods Used for Comparison** We compare Max Suppression Regularization with several variants of Label Smoothing methods, such as Zipf Label Smoothing (Liang et al., 2022) and Online Label Smoothing (Zhang et al., 2021). In cases where official implementations are available for other approaches, we adopt them directly; otherwise, we meticulously adhere to the descriptions in the respective papers for our implementations. To ensure experimental rigor and facilitate fair comparisons, all training hyperparameters are maintained identical to those of the baseline models, except for method-specific hyperparameters unique to each approach. We additionally adopt a specially designed linearly increasing $\alpha$ scheduler, which is shown to benefit the training in general, please refer to details in appendix E. It is adopted by default for both MaxSup and Label Smoothing.

#### 5.1.2 EXPERIMENT RESULTS

**Convnet Comparison** The results presented in Table 4 demonstrate the effectiveness of MaxSup regularization compared to other smoothing and self-distillation methods for training different convolutional networks on ImageNet and CIFAR100. MaxSup consistently achieves the highest accuracy among label smoothing alternatives, whereas OLS (Zhang et al., 2021) and Zipf-LS (Liang et al., 2022) fail to deliver stable performance, demonstrating that the previous empirical justification of such empirical methods is limited to certain training schemes.

In our implementation of OLS and Zipf-LS, we adhered to the methodologies and method-specific hyperparameters as outlined in their respective official codebases. However, it is important to note that we did not adopt their training recipes. For instance, the original OLS paper employs a Step Learning Rate Scheduler over 250 epochs with an initial learning rate of 0.1. Similarly, the Zipf-LS implementation utilizes 100 epochs alongside other improved training recipes.

---

[1]https://github.com/pytorch/vision

Table 4: Comparison of the performance of classic convolutional neural networks on ImageNet and CIFAR100. The training script used was consistent with TorchVision V1 Weight, but a larger batch size was employed to accelerate the experimental process. We adjusted the learning rate based on the linear scaling principle of the learning rate and batch size.

| Method | ImageNet | | | | CIFAR100 | | | |
|---|---|---|---|---|---|---|---|---|
| | Resnet-18 | Resnet-50 | Resnet-101 | MobileNetV2 | Resnet-18 | Resnet-50 | Resnet-101 | MobileNetV2 |
| Baseline | 69.11±0.12 | 76.44±0.10 | 76.00±0.18 | 71.42±0.12 | 76.16±0.18 | 78.69±0.16 | 79.11±0.21 | 68.06±0.06 |
| Label Smoothing | 69.38±0.19 | 76.65±0.11 | 77.01±0.15 | 71.40±0.09 | 77.05±0.17 | 78.88±0.13 | 79.19±0.25 | 69.65±0.08 |
| Zipf-LS$^*$ | 69.43±0.13 | 76.89±0.17 | 76.91±0.14 | 71.24±0.16 | 76.21±0.12 | 78.75±0.21 | 79.15±0.18 | 69.39 ±0.08 |
| OLS$^*$ | 69.45±0.15 | 76.81±0.21 | 77.12±0.17 | 71.29±0.11 | 77.33±0.15 | 78.79±0.12 | 79.25±0.15 | 68.91±0.11 |
| **MaxSup** | **69.59±0.13** | **77.08±0.07** | **77.33±0.12** | **71.59±0.17** | **77.82±0.15** | **79.15±0.13** | **79.41±0.19** | **69.88±0.07** |
| Logit Penalty(single run) | 66.97 | 74.21 | 75.17 | 70.39 | 76.41 | 78.90 | 78.89 | 69.46 |

Table 5: Comparison of DeiT-Small accuracy (%) with Other Label Smoothing Variants. Note that due to time limit, only the results of single runs for the setup without CutMix&Mixup are available.

| Model | Method | Acc. w/ CutMix&Mixup | | Acc. w/o CutMix&Mixup | |
|---|---|---|---|---|---|
| | | Mean | Std | run 1 | std |
| Deit-Small (Touvron et al., 2021) | Baseline | 79.69 | 0.11 | 74.21 | - |
| | Label Smoothing | 79.81(+0.12) | 0.09 | 76.12 | - |
| | Zipf-LS | 79.88(+0.19) | 0.08 | 75.48 | - |
| | OLS | 79.95(+0.27) | 0.12 | 75.98 | - |
| | **MaxSup** | **80.16(+0.47)** | **0.09** | 76.58 | - |

**Deit Comparison** Table 5 presents the performance comparison of various regularization methods applied to the DeiT-Small model on ImageNet. MaxSup demonstrates strong performance, achieving an accuracy of $80.16\%$, which surpasses Label Smoothing by $0.35\%$ points.

*Label Smoothing variants such as Zipf's and OLS show only comparable performance to standard Label Smoothing.* The marginal increase of $0.07\%$ and $0.14\%$ are statistically insignificant compared to the standard deviations, suggesting these techniques may be less effective for vision transformer architectures probably due to their heavy data augmentation pipline. These results further support the effectiveness of MaxSup across different model architectures, particularly in scenarios where other regularization techniques may struggle.

### 5.2 EXTENDED EVALUATION BEYOND IMAGE CLASSFICATION

In order to verify that MaxSup can generalize to different applications, we also evaluate our method on the task of **Machine Translation** and **Semantic Segmentation**.

**Machine Translation** We train a 12-layer Transformer model with encoder-decoder architecture (Vaswani, 2017) from scratch on the IWSLT 2014 German to English dataset (Cettolo et al., 2017), following the training setup of fairseq repository [2]. Under the same setting, we also train the transformer with MaxSup in place of Label Smoothing in the attention layers, following the common setup in previous work. The single best checkpoint and a beam size of 5 is adopted. The detokenized SacreBLEU (Post, 2018) scores of 3 runs are compared in table 6. The results demonstrate that MaxSup yields an improvement of 0.3 over baseline, which is $200\%$ relatively larger compared to the 0.1 improvement of Label Smoothing. While this enhancement may not appear substantial, it likely stems from the constraints of downstream tasks. Nevertheless, the improvement is statistically significant, as it exceeds the standard deviation.

**Semantic Segmentation** We employ the MMSegmentation framework[3] for this task. Specifically, we utilized the UperNet architecture (Xiao et al., 2018) with the DeiT-Small backbone to perform semantic segmentation on the ADE20K dataset. The backbones trained with both Label Smoothing and MaxSup on ImageNet1K are compared to the baseline. In the fine-tuning stage, the vanilla Cross-Entropy loss is used for all models. In table 7 our results show that MaxSup achieves a

---

[2]https://github.com/facebookresearch/fairseq

[3]https://github.com/open-mmlab/mmsegmentation

mean Intersection over Union (mIoU) of 44.1, outperforming the 43.7 mIoU obtained with Label Smoothing. This also supports the improved feature representation of models trained with MaxSup.

Table 6: Comparison of Label Smoothing and MaxSup on IWSLT 2014 German to English Dataset.

| Model | Param. | Method | BLEU score |
|---|---|---|---|
| Transformer(Vaswani, 2017) | 38 M | Baseline | 34.3 ± 0.09 |
| | | Label Smoothing | 34.4 (+0.1) ± 0.07 |
| | | MaxSup | **34.6 (+0.3)** ± 0.09 |

Table 7: Comparison of Label Smoothing and MaxSup on on ADE20K validation set, and the best result on ADE20K with only ImageNet-1K as training data in pretraining.

| Backbone | Segmentation Architecture | Method | mIoU(MS) |
|---|---|---|---|
| DeiT-Small (Touvron et al., 2021) | UperNet(Xiao et al., 2018) | Baseline | 43.4 |
| | | Label Smoothing | 43.7 (+0.3) |
| | | MaxSup | **44.1 (+0.7)** |

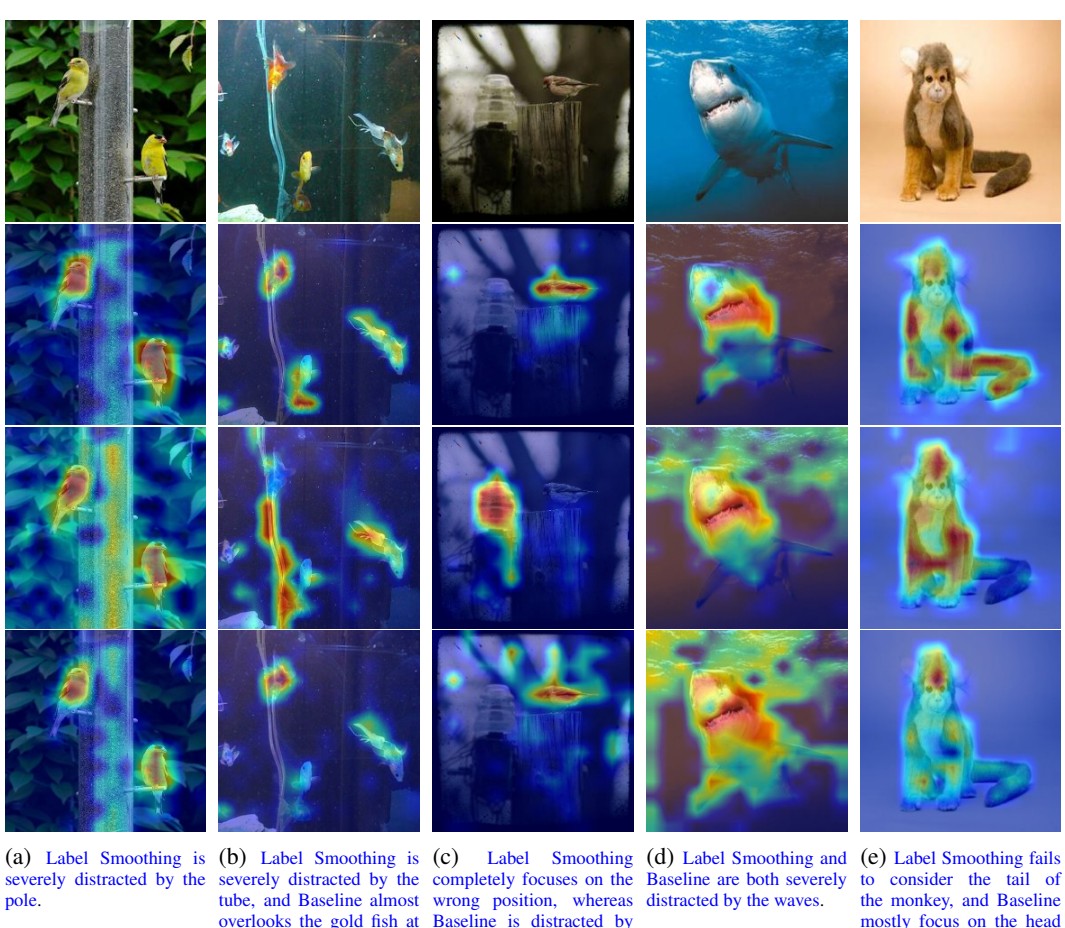

(a) Label Smoothing is severely distracted by the pole.

(b) Label Smoothing is severely distracted by the tube, and Baseline almost overlooks the gold fish at bottom.

(c) Label Smoothing completely focuses on the wrong position, whereas Baseline is distracted by the surrounding objects.

(d) Label Smoothing and Baseline are both severely distracted by the waves.

(e) Label Smoothing fails to consider the tail of the monkey, and Baseline mostly focus on the head forehead.

Figure 2: We visualize the class activation map using GradCAM (Selvaraju et al., 2019) from Deit-Small models trained with MaxSup (2nd row), Label Smoothing (3rd row) and Baseline (4th row). The first row are original images. The results show that MaxSup training with MaxSup can reduce the distraction by non-target class, whereas Label Smoothing increases the model's vulnerability to interference, causing the model partially or completely focusing on incorrect objects, due to the loss of richer information of individual samples.

## 5.3 CLASS ACTIVATION MAP

To visualize the impact of MaxSup on the model's decision-making compared to label smoothing, we adopt Gradient-weighted Class Activation Mapping (Grad-CAM), a technique by (Selvaraju et al., 2019) that generates class-discriminative localization maps. We employed the DeiT-Small to perform our experiments, comparing the models trained with MaxSup (second row), Label Smoothing (third row) and standard Cross-Entropy baseline (fourth row) in Figure 2.

As illustrated in Figure 2, the model trained with MaxSup demonstrates a clear advantage when non-target salient objects are present in the background. MaxSup reduces the model's distraction by these objects, such as the pole in the 'Bird' image, the tube in the 'Goldfish' image, and the cap in the 'House Finch' image. In contrast, the model trained with Label Smoothing often loses focus or incorrectly attends to these background objects. Figure 2a and 2b demonstrate a pattern of distraction, where the attention of the model trained with Label Smoothing is partially disrupted, although the classification remains correct. Figure 2c depicts overconfidence in incorrect samples, leading to misclassification, highlighting the negative impact of the Error-Enhancement component. Beyond the robustness to background distractions, MaxSup also improves the coverage of object features. For instance, the model trained with Label Smoothing misses important details, such as the fin in the 'Shark' image and the tail in the 'Monkey' image, both of which are successfully captured by the model trained with MaxSup. This supports our analysis in Appendix E that MaxSup better preserves the richer information of individual class samples beyond the class-specific information.

## 6 CONCLUSION

In this work, we have uncovered the underlying mechanism behind the previously identified issues in Label Smoothing and proposed MaxSup as a remedy. Our analysis reveals that Label Smoothing inherently fails to regularize the incorrect predictions and even encourages overconfidence in them, potentially hindering the model's ability to learn from challenging examples. MaxSup addresses this limitation by consistently applying the intended regularization effect during training, regardless of whether the prediction is correct or not. Our extensive analysis and experiments demonstrate that MaxSup not only improves task performance but also leads to larger intra-class variance as well as inter-class separation in the feature space over Label Smoothing. This enables the model to retain richer information of individual samples, leading to improved transfer learning. The class activation maps further support our analysis, through the more accurate localization and better coverage of class objects, as well as reduced distraction by irrelevant background objects.

**Limitation and Future Work** Müller et al. (2019) show that teachers trained on LS lead to degraded performance in Knowledge Distillation (Hinton, 2015), and Guo et al. (2024) observed accelerated convergence with LS via the conditioning number analysis. Thus it would be interesting to explore the impact of MaxSup on Knowledge Distillation and training convergence. We leave such investigations to future work.

# 7 REPRODUCIBILITY STATEMENT

The results of the code are reproducible, as detailed in Appendix D and the training setups in Section 5.1.1 and Section 5.2. We have also provided the link to the anonymous code repository for this paper in the Abstract.

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

## A   PROOF OF LEMMA 3.2

*Proof.* We aim to demonstrate the validity of Lemma 3.2, which states:

$$H(\mathbf{s}, \mathbf{q}) = H(\mathbf{y}, \mathbf{q}) + L_{LS} \tag{9}$$

where $L_{LS} = \alpha \left( H \left( \frac{1}{K}, \mathbf{q} \right) - H(\mathbf{y}, \mathbf{q}) \right)$

Let us proceed with the proof:

We begin by expressing the cross-entropy $H(\mathbf{s}, \mathbf{q})$:

$$H(\mathbf{s}, \mathbf{q}) = - \sum_{k=1}^{K} s_k \log q_k \tag{10}$$

In the context of label smoothing, $s_k$ is defined as:

$$s_k = (1 - \alpha) y_k + \frac{\alpha}{K} \tag{11}$$

where $\alpha$ is the smoothing parameter, $y_k$ is the original label, and $K$ is the number of classes.

Substituting this expression for $s_k$ into the cross-entropy formula:

$$H(\mathbf{s}, \mathbf{q}) = - \sum_{k=1}^{K} \left( (1 - \alpha) y_k + \frac{\alpha}{K} \right) \log q_k \tag{12}$$

Expanding the sum:

$$H(\mathbf{s}, \mathbf{q}) = -(1 - \alpha) \sum_{k=1}^{K} y_k \log q_k - \frac{\alpha}{K} \sum_{k=1}^{K} \log q_k \tag{13}$$

We recognize that the first term is equivalent to $(1 - \alpha) H(\mathbf{y}, \mathbf{q})$, and the second term to $\alpha H(\frac{1}{K}, \mathbf{q})$. Thus:

$$H(\mathbf{s}, \mathbf{q}) = (1 - \alpha) H(\mathbf{y}, \mathbf{q}) + \alpha H \left( \frac{1}{K}, \mathbf{q} \right) \tag{14}$$

Rearranging the terms:

$$H(\mathbf{s}, \mathbf{q}) = H(\mathbf{y}, \mathbf{q}) + \alpha \left( H \left( \frac{1}{K}, \mathbf{q} \right) - H(\mathbf{y}, \mathbf{q}) \right) \tag{15}$$

We can now identify $H(\mathbf{y}, \mathbf{q})$ as the original cross-entropy loss and $L_{LS} = \alpha \left( H \left( \frac{1}{K}, \mathbf{q} \right) - H(\mathbf{y}, \mathbf{q}) \right)$ as the Label Smoothing loss.

Therefore, we have demonstrated that:

$$H(\mathbf{s}, \mathbf{q}) = H(\mathbf{y}, \mathbf{q}) + L_{LS} \tag{16}$$

with $L_{LS}$ as defined in the lemma. It is noteworthy that the original cross-entropy loss $H(\mathbf{y}, \mathbf{q})$ remains unweighted by $\alpha$ in this decomposition, which is consistent with the statement in Lemma 3.2

## B Proof of Theorem 3.3

*Proof.* We will prove both cases of Theorem 3.3 separately.

**Without Cutmix and Mixup**

We aim to prove Equation equation 4:

$$L_{LS} = \alpha(z_{gt} - \frac{1}{K} \sum_{k=1}^{K} z_k) \tag{17}$$

Let $\mathbf{s}$ be the smoothed label vector and $\mathbf{q}$ be the predicted probability vector. We start with the cross-entropy between $\mathbf{s}$ and $\mathbf{q}$:

$$H(\mathbf{s}, \mathbf{q}) = -\sum_{k=1}^{K} s_k \log q_k \tag{18}$$

With label smoothing, $s_k = (1 - \alpha)y_k + \frac{\alpha}{K}$, where $\mathbf{y}$ is the one-hot ground truth vector and $\alpha$ is the smoothing parameter. Substituting this:

$$H(\mathbf{s}, \mathbf{q}) = -\sum_{k=1}^{K}[(1 - \alpha)y_k + \frac{\alpha}{K}] \log q_k \tag{19}$$

Expanding:

$$H(\mathbf{s}, \mathbf{q}) = -(1 - \alpha)\sum_{k=1}^{K} y_k \log q_k - \frac{\alpha}{K} \sum_{k=1}^{K} \log q_k \tag{20}$$

Since $\mathbf{y}$ is a one-hot vector, $\sum_{k=1}^{K} y_k \log q_k = \log q_{gt}$, where $gt$ is the index of the ground truth class:

$$H(\mathbf{s}, \mathbf{q}) = -(1 - \alpha)\log q_{gt} - \frac{\alpha}{K} \sum_{k=1}^{K} \log q_k \tag{21}$$

Using the softmax function, $q_k = \frac{e^{z_k}}{\sum_{j=1}^{K} e^{z_j}}$, we can express $\log q_k$ in terms of logits:

$$\log q_k = z_k - \log(\sum_{j=1}^{K} e^{z_j}) \tag{22}$$

Substituting this into our expression:

$$H(\mathbf{s}, \mathbf{q}) = -(1 - \alpha)[z_{gt} - \log(\sum_{j=1}^{K} e^{z_j})] - \frac{\alpha}{K} \sum_{k=1}^{K}[z_k - \log(\sum_{j=1}^{K} e^{z_j})] \tag{23}$$

$$= -(1 - \alpha)z_{gt} + (1 - \alpha)\log(\sum_{j=1}^{K} e^{z_j}) - \frac{\alpha}{K} \sum_{k=1}^{K} z_k + \alpha \log(\sum_{j=1}^{K} e^{z_j}) \tag{24}$$

$$= -(1 - \alpha)z_{gt} - \frac{\alpha}{K} \sum_{k=1}^{K} z_k + \log(\sum_{j=1}^{K} e^{z_j}) \tag{25}$$

Rearranging:

$$H(\mathbf{s}, \mathbf{q}) = -z_{gt} + \log(\sum_{j=1}^{K} e^{z_j}) + \alpha[z_{gt} - \frac{1}{K}\sum_{k=1}^{K} z_k] \tag{26}$$

We can identify:

- $H(\mathbf{y}, \mathbf{q}) = -z_{gt} + \log(\sum_{j=1}^{K} e^{z_j})$ (cross-entropy for one-hot vector $\mathbf{y}$)

- $L_{LS} = \alpha[z_{gt} - \frac{1}{K}\sum_{k=1}^{K} z_k]$

Thus, we have proven:

$$H(\mathbf{s}, \mathbf{q}) = H(\mathbf{y}, \mathbf{q}) + L_{LS} \tag{27}$$

**With Cutmix and Mixup**

Now we prove Equation equation 5:

$$L'_{LS} = \alpha((\lambda z_{gt1} + (1-\lambda)z_{gt2}) - \frac{1}{K}\sum_{k=1}^{K} z_k) \tag{28}$$

With Cutmix and Mixup, the smoothed label becomes:

$$s_k = (1-\alpha)(\lambda y_{k1} + (1-\lambda)y_{k2}) + \frac{\alpha}{K} \tag{29}$$

where $y_{k1}$ and $y_{k2}$ are one-hot vectors for the two ground truth classes from mixing, and $\lambda$ is the mixing ratio.

Starting with the cross-entropy:

$$H(\mathbf{s}, \mathbf{q}) = -\sum_{k=1}^{K} s_k \log q_k \tag{30}$$

$$= -\sum_{k=1}^{K} [(1-\alpha)(\lambda y_{k1} + (1-\lambda)y_{k2}) + \frac{\alpha}{K}] \log q_k \tag{31}$$

$$= -(1-\alpha)\sum_{k=1}^{K}(\lambda y_{k1} + (1-\lambda)y_{k2}) \log q_k - \frac{\alpha}{K}\sum_{k=1}^{K} \log q_k \tag{32}$$

Since $y_{k1}$ and $y_{k2}$ are one-hot vectors:

$$H(\mathbf{s}, \mathbf{q}) = -(1-\alpha)(\lambda \log q_{gt1} + (1-\lambda)\log q_{gt2}) - \frac{\alpha}{K}\sum_{k=1}^{K} \log q_k \tag{33}$$

where $gt1$ and $gt2$ are the indices of the two ground truth classes.

Using $q_k = \frac{e^{z_k}}{\sum_{j=1}^{K} e^{z_j}}$, we express in terms of logits:

$$H(\mathbf{s}, \mathbf{q}) = -(1-\alpha)[\lambda(z_{gt1} - \log(\sum_{j=1}^{K} e^{z_j})) + (1-\lambda)(z_{gt2} - \log(\sum_{j=1}^{K} e^{z_j}))] \tag{34}$$

$$- \frac{\alpha}{K}\sum_{k=1}^{K}[z_k - \log(\sum_{j=1}^{K} e^{z_j})] \tag{35}$$

Simplifying:

$$H(\mathbf{s}, \mathbf{q}) = -(1-\alpha)[\lambda z_{gt1} + (1-\lambda)z_{gt2}] + (1-\alpha)\log(\sum_{j=1}^{K} e^{z_j}) \tag{36}$$

$$- \frac{\alpha}{K}\sum_{k=1}^{K} z_k + \alpha \log(\sum_{j=1}^{K} e^{z_j}) \tag{37}$$

$$= -(1-\alpha)[\lambda z_{gt1} + (1-\lambda)z_{gt2}] - \frac{\alpha}{K}\sum_{k=1}^{K} z_k + \log(\sum_{j=1}^{K} e^{z_j}) \tag{38}$$

Rearranging:

$$H(\mathbf{s}, \mathbf{q}) = -[\lambda z_{gt1} + (1-\lambda)z_{gt2}] + \log(\sum_{j=1}^{K} e^{z_j}) \tag{39}$$

$$+ \alpha[\lambda z_{gt1} + (1-\lambda)z_{gt2} - \frac{1}{K}\sum_{k=1}^{K} z_k] \tag{40}$$

We can identify:

- $H(\mathbf{y}', \mathbf{q}) = -[\lambda z_{gt1} + (1-\lambda)z_{gt2}] + \log(\sum_{j=1}^{K} e^{z_j})$ (cross-entropy for mixed label $\mathbf{y}'$)

- $L'_{LS} = \alpha[\lambda z_{gt1} + (1-\lambda)z_{gt2} - \frac{1}{K}\sum_{k=1}^{K} z_k]$

Thus, we have proven:

$$H(\mathbf{s}, \mathbf{q}) = H(\mathbf{y}', \mathbf{q}) + L'_{LS} \tag{41}$$

This completes the proof for both cases of Theorem 3.3.

## C   GRADIENT ANALYSIS

### C.1   NEW OBJECTIVE FUNCTION

The Cross Entropy with Max Suppression is defined as:

$$L_{\text{MaxSup},t}(x, y) = H\left(y_k + \frac{\alpha}{K} - \alpha \cdot \mathbf{1}_{k=\text{argmax}(\boldsymbol{q})}, \boldsymbol{q}_t^S(x)\right)$$

where $H(\cdot, \cdot)$ denotes the cross-entropy function.

### C.2   GRADIENT ANALYSIS

The gradient of the loss with respect to the logit $z_i$ for each class $i$ is derived as:

$$\partial_i^{\text{MaxSup},t} = y_{t,i} - y_i - \frac{\alpha}{K} + \alpha \cdot \mathbf{1}_{i=\text{argmax}(\boldsymbol{q})}$$

We analyze this gradient under two scenarios:

**Scenario 1: Model makes correct prediction**

In this case, Max Suppression is equivalent to Label Smoothing. When the model correctly predicts the target class ($\text{argmax}(\boldsymbol{q}) = \text{GT}$), the gradients are:

- For the target class (GT): $\partial_{\mathrm{GT}}^{\mathrm{MaxSup},t} = q_{t,\mathrm{GT}} - \left(1 - \alpha \left(1 - \frac{1}{K}\right)\right)$

- For non-target classes: $\partial_i^{\mathrm{MaxSup},t} = q_{t,i} - \frac{\alpha}{K}$

**Scenario 2: Model makes wrong prediction**

When the model incorrectly predicts the most confident class ($\mathrm{argmax}(\boldsymbol{q}) \neq \mathrm{GT}$), the gradients are:

- For the target class (GT): $\partial_{\mathrm{GT}}^{\mathrm{MaxSup},t} = q_{t,\mathrm{GT}} - \left(1 + \frac{\alpha}{K}\right)$

- For non-target classes (not most confident): $\partial_i^{\mathrm{MaxSup},t} = q_{t,i} - \frac{\alpha}{K}$

- For the most confident non-target class: $\partial_i^{\mathrm{MaxSup},t} = q_{t,i} + \alpha \left(1 - \frac{1}{K}\right)$

The Max Suppression regularization technique implements a sophisticated gradient redistribution strategy, particularly effective when the model misclassifies samples. When the model's prediction ($\mathrm{argmax}(\boldsymbol{q})$) differs from the ground truth (GT), the gradient for the incorrectly predicted class is increased by $\alpha(1 - \frac{1}{K})$, resulting in $\partial_{\mathrm{argmax}(\boldsymbol{q})}^{\mathrm{MaxSup},t} = q_{t,\mathrm{argmax}(\boldsymbol{q})} + \alpha(1 - \frac{1}{K})$. Simultaneously, the gradient for the true class is decreased by $\frac{\alpha}{K}$, giving $\partial_{\mathrm{GT}}^{\mathrm{MaxSup},t} = q_{t,\mathrm{GT}} - (1 + \frac{\alpha}{K})$, while for all other classes, the gradient is slightly reduced by $\frac{\alpha}{K}$: $\partial_i^{\mathrm{MaxSup},t} = q_{t,i} - \frac{\alpha}{K}$. This redistribution adds a substantial positive gradient to the misclassified class while slightly reducing the gradients for other classes. The magnitude of this adjustment, controlled by the hyperparameter $\alpha$, effectively penalizes overconfident errors and encourages the model to focus on challenging examples. By amplifying the learning signal for misclassifications, Max Suppression regularization promotes more robust learning from difficult or ambiguous samples.

# D  PSEUDO CODE

We provide pseudo code to give a clearer explanation of the implementation.

---

**Algorithm 1** Gradient Descent with Max Suppression (MaxSup)

---

**Require:** Dataset $D = \{(\boldsymbol{x}^{(i)}, \boldsymbol{y}^{(i)})\}_{i=1}^N$, learning rate $\eta$, number of iterations
    $T$, regularization factor $\alpha$, a classifier $f_\theta(\cdot)$
1: Initialize the network weights $\theta$ randomly
2: **for** $t = 1$ to $T$ **do**
3:     **for** each $(\boldsymbol{x}^{(i)}, \boldsymbol{y}^{(i)})$ in $D$ **do**
4:         Compute logits: $\boldsymbol{z}^{(i)} = f_\theta(\boldsymbol{x}^{(i)})$
5:         Compute predicted probabilities: $\boldsymbol{q}^{(i)} = \mathrm{softmax}(\boldsymbol{z}^{(i)})$
6:         Compute Cross-Entropy loss: $L_{CE} = H(\boldsymbol{y}^{(i)}, \boldsymbol{q}^{(i)})$
7:         Compute Max Suppression loss: $L_{MaxSup} = z_{max} - \frac{1}{K} \sum_{k \in K} z_k$
8:         Compute the sum: $L = L_{CE} + \alpha L_{MaxSup}$
9:         Update the weights: $\theta \mathrel{-}= \eta \frac{\partial L}{\partial \theta}$
10:    **end for**
11: **end for**

---

# E  INCREASING SMOOTHING WEIGHT SCHEDULE

We hypothesize that, as the number of training epochs increases, the model improves its accuracy progressively and potentially becomes more confident about its predictions. In consequence, it might be necessary to gradually increase $\alpha$ to discourage the model's over-confidence. Therefore, we additionally propose to adopt a linearly increasing $\alpha$ schedule.

Table 8 shows the impact of a linear $\alpha$ scheduler on Label Smoothing and MaxSup. Both methods benefit from the scheduler, with LS improving from 75.91% to 76.16% and MaxSup improving from

76.12% to 76.58% with the scheduler. It can be seen that MaxSup benefits more from increasing $\alpha$ during training, with $0.46\%$ percentage point gain over baseline compared to LS's $0.25\%$. This result also supports our analysis that MaxSup fixes the inconsistent regularization and Error-Enhancement issue of Label Smoothing upon incorrect predictions.

Table 8: Effect of Alpha Scheduler. * denotes that the baseline model does not incorporate the alpha parameter, $t$ and $T$ represent the current epoch number and the total number of epochs.

| Configuration | Formulation | Accuracy | |
| --- | --- | --- | --- |
| | | $\alpha = 0.1$ | $\alpha = 0.1 + 0.1\frac{t}{T}$ |
| Baseline | - | 74.21* | |
| LS | $\alpha(z_{gt} - \frac{1}{K}\sum_{k \in K} z_k)$ | 75.91 | 76.16 |
| MaxSup | $\alpha(z_{max} - \frac{1}{K}\sum_{k \in K} z_k)$ | 76.12 | 76.58 |

subsectionVisualization of the Learned Feature Space

To visualize the difference between Max Suppression Regularization and Label Smoothing in the learned feature space, we project the feature representations from the penultimate layer into a 2D space, following Müller et al. (2019). Given three semantically similar classes, we construct an orthonormal basis for the plane intersecting their templates. We then project the penultimate layer activations of examples from these classes onto this plane. To ensure the displayability and ease of understanding of the images, we randomly sample 80 samples from the corresponding training or validation sets for the three categories separately. We select these classes based on two criteria: **1) Semantic Similarity:** Select the 3 categories that are semantically similar; **2) Confusion:** Select a class, and then find two additional classes that the model trained with Label Smoothing is most likely to confuse when predicting images of this class (fig. 3c and fig. 4c), and vice versa (fig. 3d and fig. 4d).

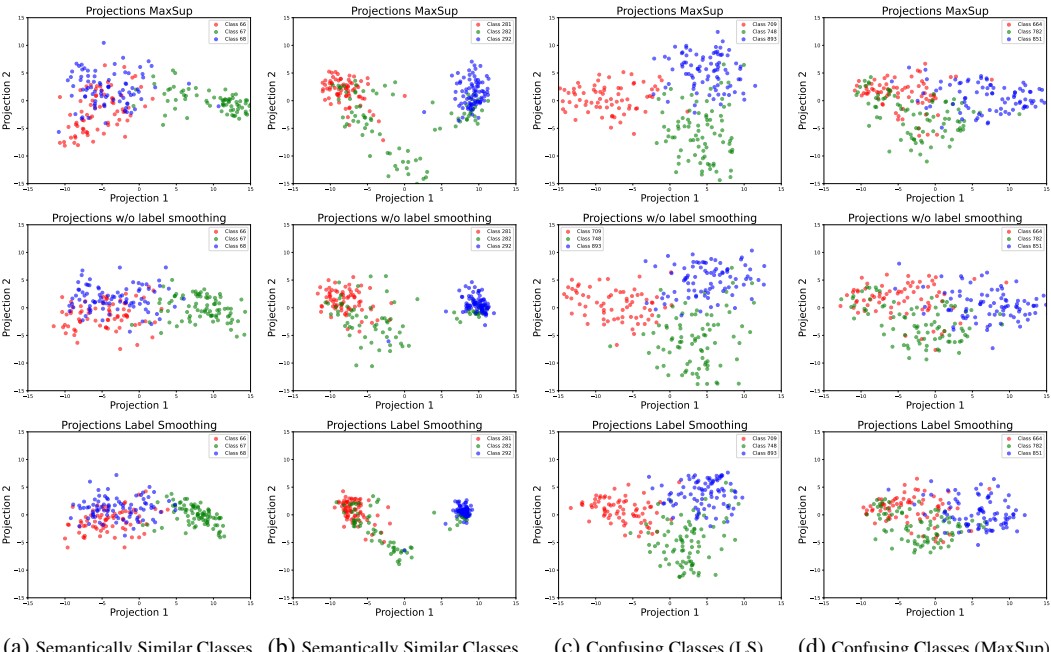

(a) Semantically Similar Classes  (b) Semantically Similar Classes  (c) Confusing Classes (LS)  (d) Confusing Classes (MaxSup)

Figure 3: Visualization of penultimate layer's activations of Deit-small (with CutMix&Mixup) for three different classes of ImageNet validation set: The first and second rows show Deit-Small trained with MaxSup and Label Smoothing, respectively. (a) 68:Schipperke, 66:Saluki, 67:Grey Fox; (b) 282:Tow Truck, 281:Pickup, 292:Unicycle; (c) 784:Jean, 709:Shoe Shop, 893:Stinkhorn. Model trained with MaxSup exhibits both improved inter-class separability and intra-class variation, indicating enhanced classification performance and representation learning.

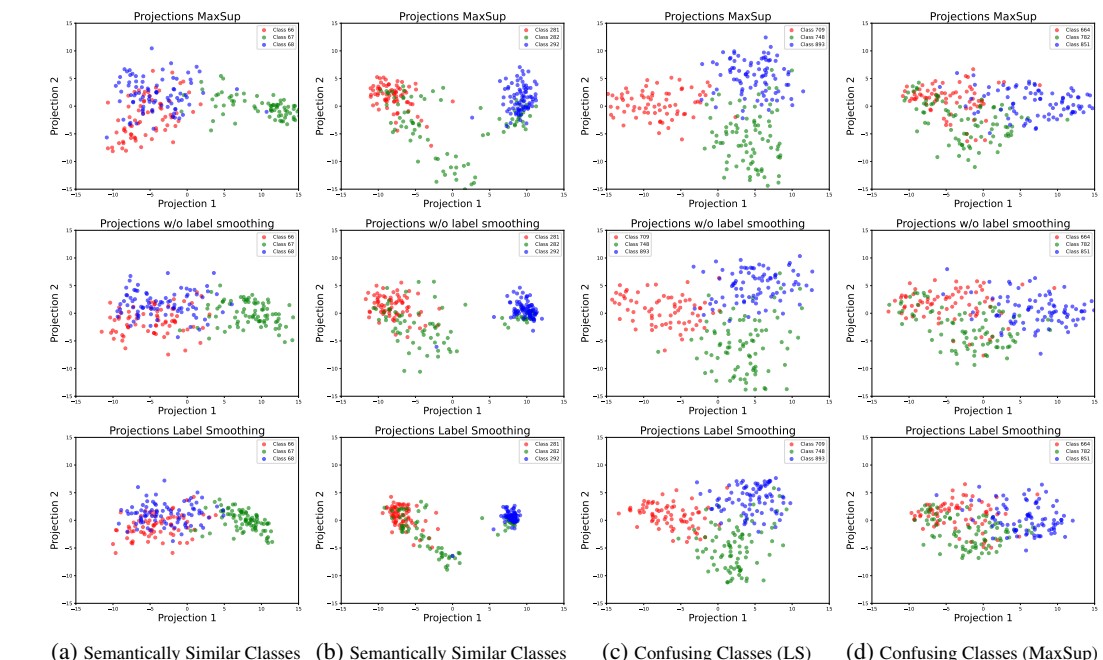

(a) Semantically Similar Classes  (b) Semantically Similar Classes  (c) Confusing Classes (LS)  (d) Confusing Classes (MaxSup)

Figure 4: Visualization of penultimate layer's activations of Deit-small (with CutMix&Mixup) for three different classes of ImageNet Train set: The first and second rows show Deit-Small trained with MaxSup and Label Smoothing, respectively. (a) 68:Schipperke, 66:Saluki, 67:Grey Fox; (b) 282:Tow Truck, 281:Pickup, 292:Unicycle; (c) 784:Jean, 709:Shoe Shop, 893:Stinkhorn. Model trained with MaxSup exhibits both improved inter-class separability and intra-class variation, indicating enhanced classification performance and representation learning.

As can be observed in Figure 4 and Figure 3, the model trained with Max Suppression has the following two major advantages against Label Smoothing:

- Improved inter-class separability: Max Suppression makes different classes more separable, indicating improved classification performance.
- Improved intra-class variation: Max Suppression better acknowledges intra-class variations, indicating improved representation learning.

For example, images of a Schipperke may differ in terms of viewpoint, lighting, or occlusion. These subtle variations are preserved in the feature space, where the semantic distances to other classes, such as Saluki or Grey Fox, adjust for each image.

