# OpenReview forum: "MaxSup: Fixing Label Smoothing for Improved Feature Representation"
_ICLR.cc/2025/Conference — Submitted to ICLR 2025_

### Official Review · Reviewer_FzkC · 2024-10-29

**Soundness:** 3
**Presentation:** 2
**Contribution:** 2
**Rating:** 5
**Confidence:** 4

**Summary:**

This paper reveals that Label Smoothing Loss can be decomposed into two opposing components: a regularization term and an error-enhancement term. Preliminary studies confirm that the performance improvements from Label Smoothing Loss are solely due to the regularization term. Based on this insight, MaxSup is proposed to eliminate the error-enhancement term for incorrect predictions. Experiments are conducted on Image Classification and Machine Translation.

**Strengths:**

1. Decomposing the Label Smoothing Loss to two terms is interesting and removing the error-enhancement term sounds reasonable.

2. The experiments show that MaxSup is better than the LS, validating the effectiveness.

**Weaknesses:**

1. The motivation of MaxSup is unclear. In Section 3.1, Label Smoothing Loss is decomposed into two components: the regularization term and the error-enhancement term. Preliminary studies validate that using only the regularization term is effective. If so, why not directly minimize the regularization term as the learning objective?

2. The incorrect prediction samples may bring noise to MaxSup during optimization.

3. Can you provide the result of using only the regularization term in Table 2 and Table 3?

4. Figure 2 shows a few samples that are correctly classified by the proposed method but incorrectly by the baseline. Are there any samples that are correctly classified by the baseline but incorrectly by the proposed method?

5. Some notations are confusing:

    a. What is $q$ in Lemma 3.2?  The definition of $q$ is found in the Appendix.

    b. Where is the $\lambda$ in Eq. (7)?

6. Some typos, e.g., Line 266, "Similarity:"

**Questions:**

See weaknesses.

---

> ### Author Response · Authors · 2024-11-23
> **Reply to reviewer FzkC**
>
> We thank the reviewer for the invaluable feedback. We address the mentioned points in the following:
>
> > **W1**: *The motivation of MaxSup is unclear. In Section 3.1, Label Smoothing Loss is decomposed into two components: the regularization term and the error-enhancement term. Preliminary studies validate that using only the regularization term is effective. If so, why not directly minimize the regularization term as the learning objective?*
>
> Replacing $z_{max}$ with $z_{gt}$ is indeed the **optimal choice**, and directly dropping the error-enhancement term is suboptimal. We have revised Section 3.1 of our manuscript to provide an extended elaboration on this aspect to improve the clarity. We would like to emphasize that **the goal of regularization is to penalize the network for being over-confident about its prediction, i.e., the top-1 position** $z_{max}$. However, Label Smoothing not only introduces an undesired error-enhancement term, but also fails to penalize the network's incorrect prediction, i.e., $z_{max} \neq z_{gt}$ in this case. Therefore, simply dropping the error-enhancement term only avoids enhancing the over-confidence in the incorrect prediction, but fails to regularize it. In contrast, replacing $z_{gt}$ with $z_{max}$ could solve both issues perfectly.
>
> This is also supported by the superior performance of MaxSup: Using the regularization term alone ($75.98\%$) only brings marginal improvement ($+0.07\%$) over Label Smoothing ($75.91\%$), from Table 1. In contrast, MaxSup ($76.12\%$) leads to larger improvement ($0.21\%$) over Label Smoothing ($75.91\%$), from Table 6 (Table 8 in the revised version). In order to improve the clarity, we have added the result of MaxSup in Table 1 in our revised manuscript.
>
>
> > **W2**: *The incorrect prediction samples may bring noise to MaxSup during optimization.*
>
> As explained in our answer to **W1**, the intended regularization, i.e., penalizes the network for being over-confident about its prediction (the peak position, i.e., $z_{max}$, should be regarded as the prediction of a classifier), can be only achieved by MaxSup when the prediction is incorrect, whereas the regularization term of Label Smoothing fails to penalize the network for being over-confident about its incorrect prediction (Inconsistent Regularization). MaxSup is also demonstrated to improve the feature representation over Label Smoothing, please refer to our answer to **Reviewer RCT5 W3**.
>
> > **W3**: *provide the result of using only the regularization term in Table 2 and Table 3*.
>
> The results of using only the regularization term in Table 2 and Table 3 are 75.84±0.18 and 79.96±0.13, which are inferior to the results of using MaxSup, i.e., 77.08±0.07 and 80.16±0.09.
>
> > **W4**: *Figure 2 shows a few samples that are correctly classified by the proposed method but incorrectly by the baseline. Are there any samples that are correctly classified by the baseline but incorrectly by the proposed method?*
>
> We measure the average top-1 accuracy on different class in ImageNet-1k dataset.
>
> | ImageNet Class ID | Diff.  | Ours Acc. | Label Smoothing Acc.|
> |--------------------|---------|---------|---------|
> | 706                | -0.04   | 0.66    | 0.70    |
> | 713                | -0.04   | 0.92    | 0.96    |
> | 714                | -0.04   | 0.78    | 0.82    |
> | 719                | -0.04   | 0.92    | 0.96    |
> | 723                | -0.04   | 0.90    | 0.94    |
> | 787                | -0.04   | 0.66    | 0.70    |
>
> **Caption**: Class-wise results.
>
> We provide three samples in butternut_squash class in validation set in ImageNet-1K.
>
>
> | Image Sample                  | Ground Truth       | Our Prediction     | Label Smoothing Prediction |
> |-------------------------|--------------------|--------------------|----------------------------|
> | Sample 1(ILSVRC2012_val_00024862.JPEG)               | butternut_squash  | head_cabbage       | butternut_squash           |
> | Sample 2(ILSVRC2012_val_00012599.JPEG)               | butternut_squash  | pineapple          | butternut_squash           |
> | Sample 3(ILSVRC2012_val_00017689.JPEG)               | butternut_squash  | bolete             | butternut_squash           |
>
> **Caption**: Sample-wise results.
>
>
> > **W5**: *Some notations are confusing.*
>
> We have resolved these issues in our revised manuscript and address your questions in the following:
>
> > W5.1: *What is $q$ in Lemma 3.2?*
>
> $\mathbf{q}$ denotes the predicted probability vector.
>
> > W5.2:  Where is the $\lambda$ in Eq. (7)?
>
> The $\lambda$ was due to sloppy writing, and has been removed.
>
> > **W6**: Some typos, e.g., Line 266, "Similarity:".
>
> We have corrected in the revised version.

---

> > ### Comment · Reviewer_FzkC · 2024-11-25
> >
> > Thanks for your reply. Here are some following questions:
> >
> > 1. I believe Line 214: The regularization term $\frac{a}{M}$ should be $\frac{a}{K}$.
> > 2. Sorry for the unclear comments. What I mean in W4 is that as the performance improvement of MaxSup is limited, there should be some samples “Correct by Baseline, Incorrect by Ours”, why only show the samples that “Correct by Ours, Incorrect by Baseline”? So, it is hard to persuade me that MaxSup is more effective than other methods. Moreover, as reviewer rCT5 and reviewer 2YeD state, these figures are difficult to interpret and understand.
> > 3. I am also looking forward to your response to the following issues 1 of Reviewer 59zm.

---

> > > ### Author Response · Authors · 2024-11-28
> > > **Reply to the follow-up questions from Reviewer FzkC**
> > >
> > > We thank the reviewer for the continued engagement and follow-up discussion. And we address the mentioned questions below:
> > >
> > > **A1**: Thanks for pointing out the typo, and we have corrected it in the revised version.
> > >
> > > **A2**: The visualization is primarily intended to demonstrate the improved **feature representation instead of prediction performance**. Specifically, it shows that MaxSup improves:
> > >
> > > - **Intra-class variation**, capturing finer-grained details within classes.
> > > - **Inter-class separability**, creating clearer distinctions between classes.
> > >
> > > Based on the suggestion of both Reviewer rCT5 and Reviewer 2YeD, we have now replaced it with **quantitative evaluation** in Table 2 and Table 3, which **clearly highlight the advantage of MaxSup against LS w.r.t feature representation and transfer learning**. Therefore, we have relocated the figure to the Appendix to make it less central to the argument. For this figure, we have removed the decision boundary to improve its clarity based on the suggestion of Reviewer 2YeD, and added an additional column using the confusing classes for MaxSup. Given the convincing quantitative evaluation, it could be entirely removed **without diminishing the significance of our work**, if the reviewer still finds it difficult to understand.
> > >
> > > As for the classification performance, we would like to kindly point out that it is **essential to consider both performance and cost** for evaluating the significance of a method. The arguably small improvements on the highly optimized baselines are substantial as they come at **almost no extra cost**: MaxSup **works as a drop-in replacement of LS**, whereas OLS has to maintain the statistics of the model prediction for each category, and Zipf employs a sophisticated pipeline to process the extra pixel-wise predictions, **introducing additional overhead** to the training process. As mentioned in our answer to Reviewer 59zm **W1** and **W2**, our MaxSup **consistently outperforms** OLS and Zipf-LS, whereas OLS and Zipf-LS **fail to deliver stable performance**. This indicates that such more sophisticated methods may rely on **additional hyperparameter tuning** to perform well.
> > >
> > > **A3**: The suggested loss by Reviewer 59zm is indeed a good example to showcase the advantage of MaxSup, and we show that it is **problematic when the network makes incorrect predictions**. As for alternative losses such as hinge loss, they simply lie **beyond the scope of this study** and **does not diminish the significance of our work**. Please refer to our answer to the following issues 1 of Reviewer 59zm for more details.

---

### Official Review · Reviewer_59zm · 2024-10-29

**Soundness:** 3
**Presentation:** 2
**Contribution:** 2
**Rating:** 5
**Confidence:** 5

**Summary:**

This paper advances the Label Smoothing technique and presents a Max Suppression method. Extensive experiments on image classification and neural machine translation tasks verify the effectiveness of the proposed method. Also, this work provides several theoretical analyses to introduce the proposed method.

**Strengths:**

1.	This work reports both CV and NLP results, which is a positive aspect.
2.	This method sounds sensible.
3.	This work presents several theorems to introduce the proposed method.

**Weaknesses:**

1.	While the method proposed in this work offer some novel ideas, the overall contribution seems to be more of an engineering refinement.

2.	The experimental results (e.g., Tables 3 and 4) do not show a substantial improvement over existing approaches.

3.	In my opinion, the primary reason for these experimental results is that only a minor enhancement was made to the Label Smoothing technique, without fundamentally overcoming its inherent limitations or proposing a more robust method. To advance in this direction, the focus should be on exploring the relationships between categories to derive more accurate soft-label encodings for each category, thereby addressing the inherent limitations of Label Smoothing.

4.	Providing two specific examples of Eq. (1) and Eq. (8) would help in understanding the difference between Label Smoothing and Max Suppression.

5.	Lines 299-302: Providing the quantitative results regarding inter-class separability and intra-class variation would be beneficial in better elucidating Figure 2.

Small issues:

1.	Line 87: calibrated classifiers M¨uller et al. (2019). -> calibrated classifiers (M¨uller et al. 2019).

2.	Which dataset was used in Figure 2?

-------------------------------------------------
Final Response

> The claim that 'there is no need to include that reference' falls outside the bounds of a rigorous scientific discussion and reflects a lack of responsibility.

Let me clarify once again: **MaxSup can be seen as an engineering enhancement for LS, while the algorithm I referenced can be viewed as an engineering enhancement for Softmax**. As this engineering technique is generally not supported by relevant academic references, there is no need to cite it. Also, they are different methods, but **their commonality lies in the fact that both are engineering tricks**. **In this context, I did not provide any misleading information**. On the contrary, **I have consistently provided corresponding responses to your feedback throughout the entire rebuttal process**. **I kindly ask that you take into account the effort I have put into this work and consider my responses objectively.** Thanks.

I acknowledge that this is an interesting work advancing the Label Smoothing (LS) method, especially in the detailed analysis of the LS method. However, I believe that the novelty of this work is not suitable for publication in ICLR.

**Questions:**

See weakness for details.

---

> ### Author Response · Authors · 2024-11-23
> **Reply to Reviewer 59zm**
>
> We thank the reviewer for the constructive feedback. We address the mentioned concerns in the following:
>
> > **W1**: The overall contribution seems to be more of an engineering refinement.
>
> We believe there exists some **misunderstanding** regarding our core contribution. Our work theoretically **reveals the reason** behind the empirically observed issue that label smoothing results in more confident errors [7]. Based on our analysis, we derive MaxSup as **the optimal solution** to such an issue. We have revised Section 3.1 of our manuscript to provide an extended elaboration on the superiority of MaxSup over Label Smoothing to improve the clarity.
>
> We would like to emphasize that **the goal of regularization is to penalize the network for being over-confident about its prediction, i.e., the top-1 position** $z_{max}$. However, Label Smoothing not only introduces an undesired error-enhancement term, but also fails to penalize the network's incorrect prediction, i.e., $z_{max} \neq z_{gt}$ in this case. And MaxSup could solve both issues perfectly by replacing $z_{gt}$ with $z_{max}$.
>
> The theoretically grounded design of MaxSup leads to its superior performance over empirical soft label approaches: From the extended evaluation of our approach using various architectures across multiple datasets in our revised manuscript, **MaxSup consistently achieves the highest accuracy among label smoothing alternatives, whereas OLS and Zipf-LS fail to deliver stable performance**, indicating that the previous empirical justification of such methods is limited to certain training schemes.
>
>
>
> | Method                | Resnet-18       | Resnet-50       | Resnet-101      | MobileNetV2     |
> |-----------------------|-----------------|-----------------|-----------------|-----------------|
> | Baseline             | 69.11±0.12      | 76.44±0.10      | 76.00±0.18      | _71.42±0.12_    |
> | Label Smoothing      | 69.38±0.19      | 76.65±0.11      | 77.01±0.15      | 71.40±0.09      |
> | ${\text{Zipf-LS}}$| 69.43±0.13      | _76.89±0.17_    | 76.91±0.14      | 71.24±0.16      |
> | ${\text{OLS}}$   | _69.45±0.15_    | 76.81±0.21      | _77.12±0.17_    | 71.29±0.11      |
> | **MaxSup**           | **69.59±0.13**  | **77.08±0.07**  | **77.33±0.12**  | **71.59±0.17**  |
>
> **Caption**: Performance comparison of classic convolutional neural networks on ImageNet-1K. The training script used was consistent with *TorchVision V1 Scripts*. Note that a larger batch size was employed to accelerate the experimental process, and the learning rate was adjusted based on the linear scaling principle.
>
>
> | Method                | Resnet-18       | Resnet-50       | Resnet-101      | MobileNetV2     |
> |-----------------------|-----------------|-----------------|-----------------|-----------------|
> | Baseline             | 76.16±0.18      | 78.69±0.16      | 79.11±0.21      | 68.06±0.06      |
> | Label Smoothing      | 77.05±0.17      | _78.88±0.13_    | 79.19±0.25      | _69.65±0.08_    |
> | ${\text{Zipf-LS}}$| 76.21±0.12      | 78.75±0.21      | 79.15±0.18      | 69.39±0.08      |
> | ${\text{OLS}}$   | _77.33±0.15_    | 78.79±0.12      | _79.25±0.15_    | 68.91±0.11      |
> | **MaxSup**           | **77.82±0.15**  | **79.15±0.13**  | **79.41±0.19**  | **69.88±0.07**  |
>
> **Caption**: Performance comparison of classic convolutional neural networks on CIFAR100. The training script used was consistent with *TorchVision V1 Scripts*.
>
>
> | Model          | Method          | Accuracy (%) |
> |----------------|-----------------|--------------|
> | Deit-Small     | Baseline        | 74.21        |
> |                | Label Smoothing | 75.91        |
> |                | Zipf-LS         | 75.48        |
> |                | OLS             | 75.98        |
> |                | **MaxSup**      | **76.58**    |
>
> **Caption:** Comparison of DeiT-Small (without CutMix&Mixup) accuracy (%) with Other Label Smoothing Variants.

---

> > ### Author Response · Authors · 2024-11-23
> > **Reply to Reviewer 59zm**
> >
> > > **W2**: *The experimental results (e.g., Tables 3 and 4) do not show a substantial improvement over existing approaches.*.
> >
> > The improvement over existing approaches in Table 3 is indeed significant: LS, Zipf-LS, and OLS improve the baseline by $0.12\%$, $0.19\%$, and $0.27\%$ respectively, while our MaxSup improves the baseline by $0.47\%$, which is **300**%, **150**%, and **74**% relative improvement to existing approaches.
> >
> > We have added the BLEU score of the baseline approach in Table 4, which is 34.3 ± 0.09. So the improvement of Label Smoothing itself is limited ($0.1\%$), probably due to the limited dataset size. In comparison, MaxSup improves over the baseline by $0.3\%$, which is a **200**% relative improvement over Label Smoothing and should be satisfactory.
> >
> > Moreover, as appreciated by **Reviewer 2YeD**, **the simplicity of MaxSup is a big plus**. In comparison, OLS [2] has to **maintain the statistics of the model prediction for each category**, and Zipf [3] employs a **sophisticated pipeline to process the extra pixel-wise predictions**, introducing additional overhead to the training process.
> >
> > > **W3**: *In my opinion, the primary reason for these experimental results is that only a minor enhancement was made to the Label Smoothing technique, without fundamentally overcoming its inherent limitations or proposing a more robust method. To advance in this direction, the focus should be on exploring the relationships between categories to derive more accurate soft-label encodings for each category, thereby addressing the inherent limitations of Label Smoothing.*
> >
> > MaxSup indeed **improves the feature representation w.r.t. class relationships in a more elegant way**, please refer to our answer to **W5**.
> >
> > In contrast, both OLS and Zipf merely demonstrate improved classification performance against label smoothing, but it is not justified whether they have mitigated the issue of Label Smoothing's semantic class relationships.
> >
> > Moreover, our work provides **theoretical justification and thoroughly derives a solution to the inherent issue of LS resulting in more confident errors**, which is empirically identified very recently [7]. Our consistently superior performance against OLS and Zipf (please refer to our answer to **W1**) also suggests that the potential of such empirical approaches could be inherently limited.
> >
> > > **W4**: *Providing two specific examples of Eq. (1) and Eq. (8) would help in understanding the difference between Label Smoothing and Max Suppression.*
> >
> >
> > We would like to kindly remind that it is more straightforward to understand the underlying mechanism of Label Smoothing and MaxSup based on our logit level reformulation of their induced loss component, as explained in our answer to **W1**. Similarly, we could understand the advantage of MaxSup against LS by **analyzing the case when the network makes an incorrect prediction** (they are equivalent when the prediction is correct) based on Eq. 1 and Eq. 8: LS encourages to **decrease the probability of the ground-truth class, even when it is supposed to be larger, and further enhances the incorrect prediction by assigning** $\frac{α}{K}$ **to the incorrect class label position**. In contrast, MaxSup encourages to decrease the probability of the incorrect top-1 prediction, and encourages to increase the probability of the ground-truth class by assigning an additional $\frac{α}{K}$ to the label position.

---

> ### Author Response · Authors · 2024-11-23
> **Reply to Reviewer 59zm**
>
> > **W5**: *Providing the quantitative results regarding inter-class separability and intra-class variation would be beneficial in better elucidating Figure 2.*
>
> We appreciate the constructive feedback and have quantitatively validated the **improved feature representation with MaxSup** over Label Smoothing using **intra-class variation** measure $d_\text{within}$ (the larger, the better transfer learning and richer similarity information—reflecting how individual examples relate to different classes) and **inter-class separability** measure $R^2=1- \frac{d_\text{within}}{d_\text{total}}$ (the larger, the better classification) metrics from [6] (suggested by **Reviewer 2YeD**), respectively:
>
> | Methods    | $d_\text{within} \uparrow$ | $d_\text{total}$ | $R^2 \uparrow$ |
> |------------|----------------------------------|-----------------------|----------------|
> | Baseline   | 0.3114                           | 0.5212                | 0.4025          |
> | LS         | 0.2632                           | 0.4862                | 0.4690          |
> | MaxSup     | 0.2926 (**+0.03**)                  | 0.6081 (**+0.12**)       | 0.5188 (**+0.05**) |
>
> **Caption:** Quantitative measures for inter-class separability and intra-class variation of feature representations, using ResNet-50 trained on ImageNet-1K. The results are calculated on the ImageNet training set. As analyzed in [6], all regularization losses reduce the relative intra-class variance in the penultimate layer representation space, but our MaxSup suffers from the least reduction of intra-class variance, significantly **improving the intra-class variance as well as the inter-class separability over Label Smoothing**.
>
>
> > **W6**: *Line 87: calibrated classifiers M¨uller et al. (2019). -> calibrated classifiers (M¨uller et al. 2019).*
>
>
> We have resolved the mentioned issue in our revised manuscript as suggested.
>
> > **W7**: *Which dataset was used in Figure 2?*
>
> It was ImageNet-1K.
>
> **reference**:
>
> - [4] Shen, Yiqing, et al. "Self-distillation from the last mini-batch for consistency regularization." Proceedings of the IEEE/CVF conference on computer vision and pattern recognition. 2022.
> - [5] Xu, Ting-Bing, and Cheng-Lin Liu. "Data-distortion guided self-distillation for deep neural networks." Proceedings of the AAAI conference on artificial intelligence. Vol. 33. No. 01. 2019.
> - [6] Kornblith et al. Why Do Better Loss Functions Lead to Less Transferable Features? NeurIPS
> 2021
> - [7] Zhu et al. Rethinking Confidence Calibration for Failure Prediction, ECCV 2022

---

> ### Comment · Reviewer_59zm · 2024-11-25
>
> Thanks for addressing my concerns in detail. I acknowledge that this is an interesting work advancing the Label Smoothing (LS) method, especially in the detailed analysis of the LS method. However, I believe the following issues still need to be addressed:
>
> 1.	The authors claim that MaxSup is the optimal solution to address LS’s two issues: (i) Error-Enhancement; and (ii) Inconsistent Regularization. However, those issues can also be tackled by the other approaches, e.g., $L = \frac{\alpha}{K} \sum\_{z\_m < z\_{gt}} (z\_{gt} – z\_m) + \frac{\alpha}{K} (z\_{\max} – z\_{gt})$. Also, many alternative solutions could be explored using other types of loss functions, e.g., hinge loss. How do you theoretically prove MaxSup is the optimal solution compared to those alternatives?
>
> 2.	Another crucial issue is that the performance improvement of MaxSup over LS seems to be limited, despite the authors' clarification. In particular, the improvement of MaxSup over LS, and even vanilla CE, is often less than 1%. If this is the first method proposed to enhance CE loss for improving generalization, such an improvement could be sufficient. However, LS is the first work that is both simple and efficient, as it does not require an **additional sorting operation** to select the top-1 logit.
>
> In summary, given the marginal performance improvement and the limited technical contribution to the LS field, I am inclined to maintain my original score.

---

> > ### Comment · Reviewer_2YeD · 2024-11-25
> >
> > I would just like to comment that I respectfully disagree with Reviewer 59zm's comments about "limited improvement". I believe that simple approaches that offer consistent improvements like MaxSup are valuable to ICLR and the machine learning community at large. I find that overly complex approaches may offer much larger boosts in specific benchmarks, but are generally less robust to changes in architecture and data modality, and are more difficult to implement, limiting their real-world value (the paper demonstrates this well when evaluating the other baselines).
> >
> > Additionally, the `max()` calculation on the model output is negligible compared to the rest of the forward and backward pass in modern deep learning.

---

> ### Comment · Reviewer_59zm · 2024-11-25
>
> Thanks for your thoughtful discussion on the “limited improvement” aspect. I appreciate your insight that simple approaches are crucial to the advancement of the machine learning (ML) community. Indeed, many simple yet effective methods have been developed over the years, significantly contributing to the progress of ML.
>
> **Label Smoothing** (LS) is a prime example of this, as it was **the first method** to address the overconfidence and hindered generalization issues caused by one-hot label encoding. While I agree that MaxSup is a simple approach, I believe that it can be seen more as a modest adjustment to LS, resulting in limited improvement, rather than a transformative solution. It addresses certain issues but does not revolutionize LS to provide a general solution to one-hot label encoding or transform the current supervised learning paradigm. Therefore, I believe that there is still much room for exploration in this field, and I hope that future work will present more groundbreaking methods to overcome the inherent limitations of LS, rather than merely repairing or patching it.

---

> ### Author Response · Authors · 2024-11-28
> **Reply to the follow-up questions from Reviewer 59zm**
>
> We are grateful to the reviewer for engaging in this follow-up discussion. And we address the mentioned aspects as follows:
>
> **A1.1**: Although the suggested loss is equivalent to MaxSup and Label Smoothing when the network makes correct predictions, it is indeed **problematic when the network makes incorrect predictions**:
>
> - For the suggested loss: the gradient w.r.t. $z_{gt}$ is $\frac{\partial L}{\partial z_{gt}}= \frac{M-1}{K}\alpha$, whereas the gradient w.r.t. $z_{max}$ is $\frac{\partial L}{\partial z_{max}}=\frac{\alpha}{K}$.
>
> - For our MaxSup: the gradient w.r.t. $z_{gt}$ is $\frac{\partial L_\text{MaxSup}}{\partial z_{gt}}= -\frac{1}{K}\alpha$, whereas the gradient w.r.t. $z_{max}$ is $\frac{\partial L_\text{MaxSup}}{\partial z_{max}}=\frac{K-1}{K}\alpha$.
>
> The **positive gradient** $\frac{M - 1}{K} \alpha$ of the suggested loss w.r.t.  $z_{gt}$  implies that the model will **decrease the already underestimated $z_{gt}$** during backpropagation. Moreover, the gradient $\frac{\alpha}{K}$  of the suggested loss w.r.t.  $z_{max}$  has a significantly smaller magnitude, which might **not sufficiently suppress the overly large $z_{max}$**. In contrast, the **negative gradient** $-\frac{1}{K}\alpha$ of MaxSup w.r.t. $z_{gt}$  encourages to **increase the underestimated $z_{gt}$**, and the significantly larger gradient $\frac{K-1}{K}\alpha$ of MaxSup w.r.t. $z_{max}$ **correctly focuses on suppressing the overly large $z_{max}$**.
>
> **A 1.2**: While exploring these alternatives might be interesting, it lies **beyond the scope of this study**: our work focuses on the issues of Cross-Entropy loss with Label Smoothing. Since Cross-Entropy is the most widely adopted loss functions in the current deep learning landscape, focusing on it **does not diminish the significance of our work**.
>
>
> **A2.1**: Regarding the reviewer's concern about the improvement in classification,  it is **essential to consider both performance and cost**  for evaluating the significance of a method. The "less than $1\%$" improvements over the highly optimized baselines are indeed substantial, **especially considering that almost no extra cost is required** (please refer to our explanation in A2.2): MaxSup **works as a drop-in replacement of LS**, whereas OLS has to maintain the statistics of the model prediction for each category, and Zipf employs a sophisticated pipeline to process the extra pixel-wise predictions, **introducing additional overhead** to the training process. As mentioned in our answer to W2, our MaxSup **consistently outperforms** OLS and Zipf-LS, whereas OLS and Zipf-LS **fail to deliver stable performance**. This indicates that such more sophisticated methods may rely on **additional hyperparameter tuning** to perform well.
>
> Moreover, it is noteworthy that our contributions are threefold, as stated clearly in our introduction section:
>
>  1) Theoretical contribution: We reveal the underlying mechanism of the previously observed defects of Label Smoothing and derive MaxSup as a remedy.
>  2) Empirical contribution: MaxSup improves classification over LS.
>  3) Empirical contribution: MaxSup improves feature representation and transfer learning over LS.
>
> In comparison, empirical soft label approaches such as **OLS and Zipf-LS only demonstrate improvements on classification over LS, but lack both theoretical contribution and demonstration of improved feature representation**.
>
>
>
>
> **A2.2**: We would like to kindly remind that **finding the top-1 logit doesn't require sorting**, and the complexity of max() operation is just $\mathcal{O}(n)$, even for large values of $n$. Furthermore, the number of classes is typically small—for instance, 1,000 in the case of ImageNet—making this computational cost **negligible** in the context of training neural networks.

---

> > ### Comment · Reviewer_59zm · 2024-11-28
> >
> > Thanks for your detailed reply. I would like to address several points that require further clarification:
> >
> > 1. I only provide an example to demonstrate that there are many methods to address the issues caused by the LS. In addition to that instance, we can also design the following loss function:
> > $L = \frac{\alpha}{K} \sum\_{\mathbf{z}\_m < \mathbf{z}\_{gt}} (\mathbf{z}\_{gt} - \mathbf{z}\_m) + \frac{K - 1}{K} \alpha \mathbf{z}\_{max} - \frac{M + 1}{K} \alpha \mathbf{z}\_{gt}$
> >
> > In this scenario, the gradients calculated by this loss function are the same as those calculated by MaxSup. **Please note that there is just one example intended to illustrate that there are many alternatives to address those issues**. Therefore, **could you provide further clarification on the claim that MaxSup is the optimal solution?** In my opinion, **the authors provide a theoretical analysis that identifies the issues of LS**, and based on this, MaxSup is proposed. However, there is a lack of theoretical analysis as to why MaxSup is the optimal solution to handle those issues.
> >
> > 2. In the industry, a common engineering trick is used to solve numerical overflow issues caused by the exponential function in the Softmax function, i.e., $Softmax(z\_i) = \frac{\exp ({z\_i – z\_{max}}) }{\sum_{j = 1}^C \exp{ (z\_j – z\_{max}}) }$. I believe that MaxSup is conceptually similar to this engineering trick. Therefore, I maintain that MaxSup is primarily an engineering trick, rather than a novel solution to revolutionizing LS.
> >
> > 3. I am sorry for my misunderstanding about the operation of max().

---

> ### Comment · Reviewer_2YeD · 2024-11-29
>
> My impression is that the authors' use of the word "optimal" was unintentionally imprecise. The paper itself doesn't make any such claims and it is difficult in and of itself to make any meaningful claims about optimality (in the sense of maximising some criterion) when we are measuring performance according to generalisation on *unseen* data.

---

> ### Author Response · Authors · 2024-12-02
> **Reply to the follow-up queries of Reviewer 59zm**
>
> We appreciate the Reviewer for **acknowledging our analysis of the issues of Label Smoothing, as well as the rationale behind the choice of MaxSup as a solution**. And we further address the questions in the following:
>
> **A1**: We thank the reviewer for suggesting alternative solutions for discussion during the rebuttal. We agree that the new example provided by the reviewer for discussion would alleviate the issues of Label Smoothing as well, since this example **follows the core design rules of MaxSup**. However, we show that MaxSup would **still be the preferable option** compared to such alternatives, due to the simplicity of MaxSup:
>
> First, the provided example is equivalent to the following:
> $$
> L = \alpha(z_{\text{max}} - \frac{1}{K} \sum_{k \in K} z_k) + \frac{\alpha}{K} \sum_{ z_{\text{gt}}<z_j<z_{\text{max}}} z_j,
> $$
> which is **essentially MaxSup (Eq. 7 in our paper) plus an additional term**.
>
> Moreover, as long as the additional term does not contain $z_{\text{max}}$ and $z_{\text{gt}}$, **a family of such losses** with the same gradients as MaxSup w.r.t. $z_{\text{max}}$ and $z_{\text{gt}}$ can be constructed:
> $$
> L = \alpha(z_{\textit{max}} - \frac{1}{K} \sum_{k \in K} z_k) + \sum_{z_j \neq z_{\textit{gt}} \text{and} z_j \neq z_{\textit{max}}} f(z_j),
> $$
> the constants such as $\alpha$ and $\frac{1}{K}$ could be further varied, but these are just trivial details and thus negligible. For such alternatives, a concern arises that **the impact of the additional term is unknown**: MaxSup **only modifies the max and gt positions** of Label Smoothing (this is more clear by comparing the soft labels in Eq. 1 and Eq. 8 in our paper), which are **essential** to solve the error-enhancement and inconsistent regularization issues of Label Smoothing. **Additional changes are avoided** (a small amount of probability mass is still uniformly distributed to the other positions as done in Label Smoothing), since they are irrelevant to the studied issues and their impact is unknown (exploration of alternatives to the uniform label component $\frac{\alpha}{K}$ in Eq. 8 is **orthogonal** to our study). In line with the widely accepted heuristic of **Occam's Razor**, i.e., **Entities should not be multiplied unnecessarily**, we assert that MaxSup is **an elegant solution**.
>
> As for the term "optimal", we totally **agree with the Reviewer** that reasoning must be provided for such a strong wording. Note that we **never claimed optimality in the paper**, and we apologize for the imprecise use of the word "optimal" in our answer to Reviewers during the rebuttal. **As pointed out by Reviewer 2YeD in the reply to Reviewer 59zm, it is difficult in and of itself to make any meaningful claims about optimality (in the sense of maximising some criterion) when we are measuring performance according to generalisation on unseen data**.  And our contributions are **independent of the assumption of optimality. Therefore, "optimality" shall be less relevant to** the evaluation of our paper.

---

> ### Author Response · Authors · 2024-12-02
> **Reply to the follow-up queries of Reviewer 59zm**
>
> **A2**: We thank the Reviewer for the explicit explanation of the remaining concern, and we would like to show that MaxSup and the mentioned engineering trick are **substantially different** from each other regarding both **concepts and mechanisms**. We believe the Reviewer's claim might simply stem from a misunderstanding of our method, and we are glad to provide a detailed explanation:
>
> - **The concepts are different**: MaxSup is **a regularization method** for penalizing overconfident predictions of a classification network, whereas the mentioned engineering trick is essentially **a normalization method** for preventing numerical overflow of Softmax function.
>
> - **The mechanisms are different**: The mentioned engineering trick is based on the **translation invariance of Softmax function**, i.e., $$Softmax_i(\mathbf{z}+\mathbf{c}) = \frac{\text{exp}(z_i + c)}{\sum_{j=1}^C \text{exp}(z_j + c)}= \frac{\text{exp}(z_i)\text{exp}(c)}{\sum_{j=1}^C \text{exp}(z_j)\text{exp}(c)} =\frac{\text{exp}(z_i)}{\sum_{j=1}^C \text{exp}(z_j)}=Softmax_i(\mathbf{z}) ,$$
>     **where $\mathbf{c}=(c,...,c)$ could be an arbitrary constant vector, which will be eliminated**. Setting $c=-z_{\text{max}}$ normalizes the maximum exponentiated value to $e^0=1$, but the output remains unchanged.
>
>     In contrast, the training objective of a classifier with MaxSup is: $$L = H(\mathbf{y}, \mathbf{q}) + \alpha(z_{\textit{max}}  - \frac{1}{K} \sum^{K}_{k = 1} z_k),$$
>     where $\mathbf{q}$ denote the predicted probability vector, and $\mathbf{y}$ denote the one-hot ground-truth label. Note that our MaxSup is **a stand-alone regularization term** in addition to the vanilla cross-entropy loss, which is **not added to the input of softmax** and can **not be eliminated**.
>
> To avoid such a misunderstanding for future readers, we carefully inspected our manuscript and suspect that it probably occurs because we only presented the stand-alone Max Suppression Loss in Eq. 7, and **we plan to improve the clarity by replacing it with the full training objective as above in our camera ready version**.
>
> Also, we would be glad if the reviewer can provide a specific reference for the heuristic so that we can add a short discussion to the paper.

---

> ### Comment · Reviewer_59zm · 2024-12-03
>
> Thank you for your detailed response. I would like to take this opportunity to clarify the following points:
>
> First, **I appreciate your acknowledgment that the proposed MaxSup may not be a theoretically optimal solution to the issues of LS**. Accordingly, my main point remains that while the authors have provided a theoretical analysis highlighting the issues of LS, there is a lack of theoretical proof for why MaxSup is presented as the optimal solution. **Given that MaxSup has not been theoretically proven to be optimal, I regard it primarily as an intuitive engineering design, making minor adjustments to LS, resulting in a modest performance improvement**. As such, I believe that MaxSup represents more of an engineering approach than an innovative solution. **It does not fundamentally revolutionize LS, but rather serves as a corrective measure**. Consequently, **the novelty and contributions of MaxSup are limited**.
>
> Second, regarding my previous example of an engineering technique, I intended to illustrate that your method should be seen as an engineering solution rather than a novel theoretical breakthrough. **I kindly request that we refrain from focusing on the differences between these methods**. I fully recognize that they are distinct approaches, but **their commonality lies in the fact that both are engineering tricks**.
>
> Third, **to avoid any further misunderstandings, I will refrain from providing any additional examples this time**.

---

> ### Author Response · Authors · 2024-12-03
>
> We apologize if we misread your previous comment. Pointing out the different effects of the regularization and normalization approaches was meant to provide clarification on the contribution of MaxSup. Again, if you would **provide us with a reference**, we gladly refer to the work in our related work section.

---

> ### Comment · Reviewer_59zm · 2024-12-03
>
> That approach to modifying softmax is simply an engineering solution, and I am not aware of a specific reference for it. Additionally, **as mentioned in my previous comment**, there is no need to include that reference. Thanks.

---

> ### Author Response · Authors · 2024-12-04
>
> We sincerely appreciate the time and effort the Reviewer has dedicated to the review process. We fully understand that, given the workload and tight time constraints, misunderstandings may occasionally occur. However, we would like to respectfully address what we believe to be an **unfounded** or **irresponsible** response.
>
> - The claim that "Given that MaxSup has not been theoretically proven to be optimal, I regard it primarily as an intuitive engineering design" is **unfounded**. Instead of proving MaxSup as an optimal solution, we have provided theoretical proof (please refer to the gradient analysis in Appendix C) that MaxSup is an **effective and concise solution** to the studied issues of Label Smoothing. Note that **even Label Smoothing and its alternatives, including Zipf-LS and OLS, are not proven as an optimal solution**, yet Label Smoothing is widely applied in training classifiers.
>
> - The claims that "regarding my previous example of an engineering technique, I intended to illustrate that your method **should be seen as an engineering solution** rather than a novel theoretical breakthrough" and "their commonality lies in **the fact that both are engineering tricks**" are **unfounded**, as the reviewer admitted "I fully recognize that they are **distinct approaches**" and "I am **not aware of** a specific reference for it".
>
> - The claim that 'there is **no need** to include that reference' falls **outside the bounds of a rigorous scientific discussion** and reflects a **lack of responsibility**.

---

### Official Review · Reviewer_2YeD · 2024-11-03

**Soundness:** 3
**Presentation:** 3
**Contribution:** 3
**Rating:** 8
**Confidence:** 4

**Summary:**

This paper analyses the effect of label smoothing for image classification. By analysing the loss they find that label smoothing  (compared to vanilla cross entropy) encourages the model to be more confident when it is incorrect. Based on this finding they propose MaxSup, a similarly simple regularisation approach that penalises the max logit rather than the logit corresponding to the ground truth label. They then demonstrate consistent improvements over label smoothing in terms of test accuracy on a range of experiments.

**Strengths:**

- The paper presents interesting analysis into a deeper understanding of label smoothing, which I appreciate. This understanding is then leveraged to propose a new approach.
- The proposed approach is very simple, and seems to perform well empirically, making it a strong candidate for a drop-in replacement/improvement on label smoothing. The **simplicity is a big plus**, as it has the potential for wider adoption and impact. Moreover, the supporting analysis seems to be solid, reducing the likelihood of the performance failing to generalise beyond the experiments in the paper.

**Weaknesses:**

Although I have tried my best to cover the content in enough detail, due to time constraints, I may have misunderstood/missed aspects of the paper. I encourage the authors to correct/clarify this if it is the case. I also welcome the authors to challenge me if they disagree with some of the more subjective points I make. I will actively engage during the rebuttal period.

1. Positioning relative to existing/concurrent work. Overall, the paper lacks appropriate acknowledgement to existing work relating to label smoothing.
    1. The empirical notion that label smoothing results in more confident errors has been around since 2022 [1], whilst the authors claim this to be a completely novel revelation. Moreover, the type of analysis performed in this paper is also present in [2]. Although [2] is not peer reviewed, and addresses a different problem, it contains substantial overlap with section 3 of the paper. In particular how the logits are suppressed differently depending on whether a prediction is likely to be correct or not. Note that I do not suspect any plagiarism, however, as existing/concurrent work, the above references should be clearly acknowledged and contextually placed in relation to this work. Although it is somewhat grey, I would deem [2] to be concurrent work as it is not published, but has been available on arxiv since March of this year. The authors should accordingly temper any claims of  "previously unknown/novel".
    2. The authors also do not reference a number of relevant background work investigating label smoothing in the context of knowledge distillation [3,4,5], which are particularly concerned with the "erasure of knowledge". [6], which empirically shows label smoothing degrades transfer learning and also empirically measures increased tightness and separation of feature clusters, is an especially important reference but is also missing. Similarly, works that investigate the effect of LS on neural collapse also measure these empirical properties [7,8]. The authors should add these papers to the related work. [6] also examines a regulariser that minimises the l2-norm of the logits (logit penalty), which is conceptually similar to MaxSup since the l2-norm tends to be dominated by the max logit (or MaxSup is like minimising the infinity-norm). Ideally, this would be added to Tabs. 2 and 3 although at the least it should be clearly referenced and discussed in the paper.
2. Lacking analysis. Although it is somewhat intuitive that encouraging confidence on errors would hurt generalisation, the additional analysis that is performed is somewhat lacking.
    1. The authors analyse the inter-class separability and intra-class variation of feature representations, however, they use only the illustrative plots found in [9], which are not very scientific. Since [9] was published in 2019 there have been a number of works that directly quantify separability/variation (e.g. [6,7]). It would greatly strengthen this part of the work if the authors were to provide such numerical measures over the training and test datasets. Moreover, there is a lack of comparison to vanilla CE, which should be the real baseline.
    2. The discussion in Sec. 3.3 is somewhat handwavy. As the visualised data is primarily *training* data, better class separation doesn't necessarily mean better accuracy on *test* data. There is also no reference/experiment to support the claim that greater intra-class variation leads to improved representation learning. The authors should use [6] as a reference in the context of these claims, as in [6] they indeed show empirical results similar to the authors claims by comparing the test accuracy vs transfer linear probing (representation learning) performance of label smoothing. Additionally, the submission would be strengthened with a transfer learning experiment (like in [6] Tab. 1). It would be great if it can be shown that MaxSup is able to improve test accuracy without sacrificing transfer performance (unlike label smoothing). The semantic segmentation experiment seems to do this, however, it is unclear whether or not the downstream segmentation loss also includes MaxSup/LS and the experiment lacks a vanilla CE baseline (which the authors should include).
    2. It is not really clear how the behaviour in Sec 3.3 follows from the analysis in 3.1/3.2. The authors ideally should be able to link the theory with regards to the different training losses to the difference in observed separation/variation. This, again, would greatly improve Sec 3.3.
    2. The paper would benefit from some additional explicit discussion on how increasing confidence on training errors would lead to worse generalisation in terms of top 1 test accuracy. Although it seems to make sense superficially, in my mind there is still a gap in reasoning. For example adding a sentence like: LS weakens the strength of supervision to the GT class on errors, when a model really should be learning to correct itself rather than being regularised to prevent overfitting.
3. Presentation, missing information and clarity.
    1. The alpha schedule in Tab. 6 comes out of nowhere. It seems to be referenced after Eq. 7 even though it does not appear there. This seems to just be sloppy writing and needs to be clarified properly. Although Tab. 6 suggests that an alpha schedule is generally beneficial, it is orthogonal to the main body of the paper. I think the paper would benefit either from an expanded (and separate) discussion and analysis on the alpha schedule, or just simply from its removal from the paper.
    2. The wording and notation in Sec. 3.1 is confusing. From line 124 onwards the use of "cross entropy", "label smoothing" and "loss" are combined in different ways to refer to distinct things that are easily confused. For example, the "cross entropy loss with label smoothing" is not the same thing as the "label smoothing loss" or the "cross entropy between two distributions" but all are related and interlinked. I would like to see the authors improve the clarity of Sec 3.1 with updated wording and maybe some additional underbraces to the equations (like in Eq. 6). Sec 3. could also benefit from an illustrative figure for LS vs MaxSup (e.g. using a bar chart to represent logits), although this is not essential.
    3. Fig. 2. is difficult to interpret. It is hard to compare when it is not clearly labelled on the figure which row is LS and which is MaxSup. The decision boundaries arguably don't contribute to the takeaways so the diagrams may be easier to interpret if they are omitted. The choice of showing test and training data together on the same plot without easily being able to tell the difference without pixel peeping is also poor (and this also doesn't contribute to the takeaways about variation/separation). Finally, I am generally lukewarm on these plots (as mentioned in 2.) as their axes can easily be linearly scaled to visually tell different stories and it is hard to normalise for this scaling (for example Muller [9] scale their axes but the authors in this paper do not). As mentioned before, numerical measures [6,7] may convey the point more clearly.
    3. Although I like the CAM figures, the authors should add a comparison to the baseline cross entropy model as well. The figures would also benefit from subcaptions describing what to look out for, e.g. "label smoothing fails to consider the tail of the monkey".
    4. Overconfidence in the context of model calibration and overconfidence in the sense of error enhancement are not clearly delineated. LS often reduces the former [9,10], but in this paper the authors show that LS increases the latter. The authors should make clear the difference between the two definitions of overconfidence to avoid confusion. This is important since LS is widely known to reduce calibration overconfidence [9,10].
    5. Generally speaking, there are a number of grammar and spelling errors. The authors should use an automated grammar checking tool to correct these.
4. Choice of Deit as a model. This choice of model as a baseline to perform ablations and analysis is odd. Deit uses hard label distillation on an additional token during training. Although this doesn't necessarily conflict with the theory presented in the paper, it does add unnecessary complexity to the experiments in my opinion. Additionally, it is unclear when and where Mixup/Cutmix are used. To me it makes a lot more sense to perform a simple analysis of vanilla CE vs label smoothing, rather than also throwing in all the additional label augmentations that come with Deit by default that may muddy the results (cutmix, mixup, hard-label distillation). Although this is a comparatively minor complaint, I would prefer it if the authors simply used ResNet-50 for Tab. 1 and Fig. 2.. Alternatively, the authors can add the Deit distillation into the theory/discussion of Sec 3.

I'd like to note that I think the **core of the paper is strong**, especially the proposed MaxSup, and am **keen to increase my score**. However, I believe that there are many issues with the manuscript in its current state that need to be addressed before it is ready for publication. I look forward to updates from the authors. Hopefully, the paper will improve via the review process.

**Edit: after extensive discussion with the authors and a substantially revised manuscript, I believe the submission has improved considerably and I am happy to raise my score from 5 to 8.**

**Edit 2: I will leave an additional comment here as it is past the deadline for discussion. I believe I have found a potential flaw in MaxSup, that exists outside of the scope of this paper (and thus may be addressed in future work). In the case of soft training targets (e.g. knowledge distillation) MaxSup may lead to potential instability. For example, for a uniform target presented by a teacher model, the penalisation of the max logit may prevent the student from stably also producing a uniform softmax output. The constant downward gradient on the max logit may jump around as the max logit varies as the softmax fluctuates locally around the uniform target. This does not affect my score as the paper does not consider training with soft targets, however, it does point to potential space for improvement in the future.**

[1] Zhu et al. Rethinking Confidence Calibration for Failure Prediction, ECCV 2022

[2] Xia et al. Understanding Why Label Smoothing Degrades Selective Classification and How to Fix It, 2024

[3] Yuan et al. Revisiting Knowledge Distillation via Label Smoothing Regularization, CVPR 2020

[4] Shen et al. Is Label Smoothing Truly Incompatible with Knowledge Distillation: An Empirical Study, ICLR 2021

[5] Chandrasegaran et al. Revisiting Label Smoothing and Knowledge Distillation Compatibility: What was Missing? ICML 2022

[6] Kornblith et al. Why Do Better Loss Functions Lead to Less Transferable Features? NeurIPS 2021

[7] Zhou et al. Are All Losses Created Equal: A Neural Collapse Perspective, NeurIPS 2022

[8] Guo et al. Cross Entropy versus Label Smoothing: A Neural Collapse Perspective, 2024

[9] Muller et al. When Does Label Smoothing Help?

[10] Liu et al. The Devil is in the Margin: Margin-based Label Smoothing for Network Calibration, CVPR 2022

**Questions:**

See weaknesses

---

> ### Author Response · Authors · 2024-11-23
> **Reply to Reviewer 2YeD**
>
> We thank the reviewer for the knowledgeable review and invaluable suggestions. We address the mentioned points in the following:
>
> > **W1**: *Positioning relative to existing/concurrent work*.
>
> We address the discussions in the following:
>
> > W1.1. *Existing work [1] empirically identified the more confident errors caused by Label Smoothing*.
>
>    We have revised our contribution as "uncovering the underlying mechanism of the recently observed defect of Label Smoothing" to acknowledge their contribution in our revision. Indeed, we regard the existence of the empirical finding as rather positive, because **our theoretical analysis reveals the reason behind their empirical finding**, which supports the validity of our theoretical analysis in return.
>
> > W1.2. *Concurrent work [2] similarly shows how the logits are suppressed differently depending on whether a prediction is likely to be correct or not*.
>
>    The observation in [2] is made by **analyzing the gradients w.r.t. logits**, while our analysis is based on **logit-level reformulation of the training objective**. The findings from both studies can be seen as mutually validating. However, **our decomposition offers an additional advantage**, as it allows us to derive MaxSup as a direct solution to the observed problem, whereas [2] merely recommend an existing post-hoc technique as remedy, i.e., logit normalization [11]. We have added this discussion in Section 3.1 in our revised manuscript.
>
> > W1.3 *Related work [3, 4, 5, 6, 7, 8] are missing*.
>
>    We have included [3-8] in the related work in our revised manuscript. Following the reviewer's suggestion, we have also provided quantitative evaluation of feature separability/variation and transfer learning performance, following [6].

---

> ### Author Response · Authors · 2024-11-23
> **Reply to Reviewer 2YeD**
>
> > **W2**:  *Lacking analysis*.
>
> We address the suggestions in the following:
>
> > W2.1: *Lack of quantitative measures over the training and test datasets, as well as comparison to vanilla CE*.
>
> We appreciate the insightful feedback and have quantitatively evaluated the intra-class variance $d_\text{within}$ and inter-class separation $R^2$ from [6], respectively:
>
>
> | Methods        | $d_\text{within} \uparrow$ | $d_\text{total}$ | $R^2 \uparrow$ |
> |----------------|----------------------------------|------------------------|---------------------------------------------------------------------|
> | Baseline       | 0.3114                           | 0.5212                 | 0.4025                                                             |
> | LS             | 0.2632                           | 0.4862                 | 0.4690                                                             |
> | MaxSup         | 0.2926 (**+0.03**)                 | 0.6081 (**+0.12**)       | 0.5188 (**+0.05**)                                                   |
> | Logit Penalty  | 0.2840                           | 0.7996                 | 0.6448                                                             |
>
> **Caption:** Quantitative measures for inter-class separability and intra-class variation of features representations, using ResNet-50 trained on ImageNet-1K. The results are calculated on the ImageNet training set.
>
> As analyzed in [6], all regularization losses reduce the relative intra-class variance in the penultimate layer representation space,  but our MaxSup suffers from the least reduction of intra-class variance, significantly improving the intra-class variance as well as the inter-class separability over Label Smoothing.
>
>
>
> > W2.2: *The claims should be supported by comparing the test accuracy vs transfer linear probing (representation learning) performance, following [6]. It is unclear whether or not the downstream segmentation loss also includes MaxSup/LS and the experiment lacks a vanilla CE baseline (which the authors should include).*
>
>
> | Methods          | Linear Transfer *val. acc* |
> |------------------|---------------------------|
> | Baseline         | 0.8143                    |
> | Label Smoothing  | 0.7458                    |
> | Logit Penalty    | 0.7242                    |
> | MaxSup           | 0.8102 (+0.0644)          |
>
> **Caption:** Validation performance of different methods based on multi-nominal Logistic Regression with $l_2$ regularization in CIFAR10 validation set. We searched the strength of the regularization from $1e-4$ to $1e2$, the search step size is increasing by an order of magnitude.
>
> We evaluated the linear transfer performance of a ResNet50 model, pre-trained on ImageNet, on the CIFAR-10 dataset using different methods following [6]. As for the segmentation task, the vanilla Cross-Entropy loss is used for fine-tuning. We have also included the result of using vanilla CE baseline in the revised manuscript as follows:
>
> | Methods          | mIoU (MS) |
> |------------------|---------------------------|
> | Baseline         | 43.4                  |
> | Label Smoothing  | 43.7 (+0.3)                   |
> | MaxSup           | 44.1  (+0.7)        |
>
> **Caption**: Comparison of Label Smoothing and MaxSup on on ADE20K validation set, and the best
> result on ADE20K with only ImageNet-1K as training data in pretraining.
>
> > W2.3: *The differences in observed separation/variation in section 3.3 should be linked to the theoretical analysis in section 3.1 and 3.2*.
>
> We would like to emphasize that **the goal of regularization is to penalize the network for being over-confident about its prediction, i.e., the top-1 position** $z_{max}$. However, Label Smoothing not only introduces an undesired error-enhancement term, but also fails to penalize the network's incorrect prediction, i.e., $z_{max} \neq z_{gt}$ in this case. Since MaxSup consistently regularizes both the correct and incorrect predictions (top-1 probability), thereby it leads to even larger inter-class separability. Moreover, MaxSup eliminates the error-enhancement defect of Label Smoothing, which may be the cause of the severely reduced intra-class variance.
>
> > W2.4: *Additional explicit discussion on how increasing confidence on training errors would lead to worse generalization in terms of top 1 test accuracy. For example adding a sentence like: LS weakens the strength of supervision to the GT class on errors, when a model really should be learning to correct itself rather than being regularized to prevent over-fitting*.
>
>
> We have revised Section 3.1 of our manuscript to provide an extended elaboration on this aspect to improve the clarity. Please also refer to our  answer to **W2.3** for more details.

---

> ### Author Response · Authors · 2024-11-23
> **Reply to Reviewer 2YeD**
>
> > **W3**: *Presentation, missing information and clarity*.
>
>
> We address the suggestions in the following:
>
> > W3.1: *The paper would benefit either from an expanded (and separate) discussion and analysis on the alpha schedule, or just simply from its removal from the paper.*
>
> Based on the reviewer's suggestion, we have revised the manuscript by removing the details of the alpha scheduler from the main text and adding a  separate section in the appendix to discuss it.
>
> > W3.2: *Improve the clarity of Sec 3.1 with updated wording and maybe some additional underbraces to the equations (like in Eq. 6).*
>
> To avoid the confusion, we have now adopted "Cross-Entropy Loss with Hard Label" and "Cross-Entropy Loss with Soft Label" to denote the vanilla training objective and the training objective with Label Smoothing, respectively. "Label Smoothing Loss" is kept to denote the the loss component introduced by Label Smoothing. In addition, we have added underbraces to each of these terms in Eq. 2 for improved clarity.
>
> > W3.3: *Omit the decision boundaries and numerical measures [6,7] may convey the point more clearly than the plots.*
>
>
> We have put Figure 2 in the appendix and will omit the decision boundaries as suggested. Instead, we provided numerical measures as mentioned in our answer to **W2**.
>
> > W3.4: *For CAM figures, the authors should add a comparison to the baseline cross entropy model as well. The figures would also benefit from subcaptions describing what to look out for, e.g. "label smoothing fails to consider the tail of the monkey".*
>
>
> We have added one row for baseline model to the CAM figures, and also improved the caption as suggested.
>
>
> > W3.5: *The authors should make clear the difference between overconfidence in the context of model calibration and overconfidence in the sense of error enhancement.*
>
> In our paper, the "overconfidence" consistently regards **the network's confidence in its top-1 prediction**, regardless of whether the prediction is correct or not. As explained in our revised analysis in our answer to **W2.3**, LS indeed only reduces the overconfidence in the correct prediction, but fails to reduce the overconfidence in the incorrect prediction (**Inconsistent Regularization**) and even further enhanced it (**Error Enhancement**).
>
> > W3.6: *Generally speaking, there are a number of grammar and spelling errors. The authors should use an automated grammar checking tool to correct these.*
>
>
> We have addressed the grammar issues as per the reviewer's suggestions.
>
> > **W4**: *Suggestion to use a clean baseline without additional label augmentations such as hard-label distillation, cutmix and mixup.*
>
>
> We would like to kindly remind that the widely adopted Deit baseline is trained on ImageNet1K from scratch, without using Knowledge Distillation. However, we agree that using a clean baseline without cutmix and mixup is a better choice. Therefore, we have added the comparison to other approaches using Deit baseline without cutmix and mixup.
>
> | Model          | Method          | Accuracy (%) |
> |----------------|-----------------|--------------|
> | Deit-Small     | Baseline        | 74.21        |
> |                | Label Smoothing | 75.91        |
> |                | Zipf-LS         | 75.48        |
> |                | OLS             | 75.98        |
> |                | **MaxSup**      | **76.58**    |
>
> **Caption:** Comparison of DeiT-Small (without CutMix&Mixup) accuracy (%) with Other Label Smoothing Variants.
>
> **Reference**:
> - [11] Luís Felipe Prates Cattelan and Danilo Silva. How to fix a broken confidence estimator: Evaluating post-hoc methods for selective classification with deep neural networks. In The 40th Conference on Uncertainty in Artificial Intelligence, 2024

---

> > ### Comment · Reviewer_2YeD · 2024-11-23
> >
> > Thanks for the updates, they look promising!
> > I will try my best to get back to the authors before the end of the discussion period (although I am busy this weekend).

---

> > > ### Author Response · Authors · 2024-11-23
> > > **Reply to Reviwer 2YeD**
> > >
> > > Thank you very much for your prompt reply! We look forward to further discussions with you.

---

> > > > ### Comment · Reviewer_2YeD · 2024-11-25
> > > >
> > > > First of all, I'd like to thank the authors for carefully addressing my review. I acknowledge that my review may have presented somewhat of a burden and appreciate the authors for making the effort to address my concerns.
> > > >
> > > > The paper has *improved considerably* with this revision. However, I still have a few (minor) points I'd like to follow up on. Hopefully the authors can reply in time for the end of the discussion period.
> > > >
> > > > **Follow up queries**
> > > >
> > > > 1. Table 2 is missing the L2 logit penalty. It is also not discussed in relation to MaxSup in Sec. 4. The ImageNet accuracy of L2 logit penalty is also missing.
> > > > 2. It doesn't appear that vanilla CE being used for downstream semantic segmentation is clarified in the text.
> > > > 3. I am still not convinced by the authors' explanation of why MaxSup may lead to the (now quantitively observed) variation and separation behaviour compared to LS. In fact, the training minimum (perfectly fitting the data, 100% training accuracy) for both cases is still a state of neural collapse where variation is minimised and separation is maximised. My gut says that the authors may be able to find an explanation by considering the neural collapse literature and applying their gradient analysis to how MaxSup and LS progress differently towards the training minimum. However, for now it remains a weakness. Given the limited time, if the authors cannot update their explanation I would prefer if they presented Sec. 4 as purely empirical.
> > > > 4. It is still not clear where and when the $\alpha$ schedule is used (is it used for both LS and MaxSup? Table. 5 comparing to Table. 8 suggests otherwise.). Please make this clear in the paper. If the $\alpha$ schedule is used for MaxSup, then it should also be used for LS, since this schedule is orthogonal to either.
> > > > 5. The captions in Fig. 2 are poorly formatted, maybe reducing the fontsize would help.
> > > > 6. I would still add a short sentence clarifying that the overconfidence discussed in this paper is not the same as that in model calibration, since LS is widely know to successfully reduce overconfidence in calibration, which may lead to reader confusion.
> > > > 7. I apologise, after inspecting the Deit codebase, it does appear that Deit-small is not trained with distillation. However, as distillation features as a prominent contribution of the Deit paper, I think the authors should add a brief sentence clarifying that they do not use distillation to avoid confusion like mine for future readers.

---

> > > > > ### Comment · Reviewer_2YeD · 2024-11-27
> > > > >
> > > > > Although the discussion period has been extended, the authors soon will be unable to update the submission pdf. Although I would very much like to give this paper up to an 8, I am right now inclined to a 6. I feel like the points raised in my previous comment (especially 1,4) prevent me from committing to a higher score. If the authors are unable to revise the pdf in time, I am still open to improving my score if they clarify in the comments and promise to revise the camera ready.
> > > > >
> > > > > Additionally, I would suggest the authors adjust their language here.
> > > > >
> > > > > >  the goal of regularization is to penalize the network for being over-confident about its prediction, i.e., the top-1 position
> > > > >
> > > > > Although I understand what you mean, it may be better to spell it out more clearly (this may help convince some of the other reviewers). Something along the lines of:
> > > > >
> > > > > "The goal of regularisation is to prevent a model from overfitting to the training datapoints. In the case of classification this may occur when a model is highly confident on the ground truth  training label. However, overfitting is not occurring when a model is incorrect as it is inherently not fit to the training data. By penalising the ground truth logit and enhancing the error on incorrect predictions, label smoothing does not prevent overfitting and instead worsens the learning on poorly fit/incorrect samples. MaxSup addresses this deficiency by instead penalising the max logit, which both penalises overfitting on well-fit datapoints (GT=prediction), as well as encouraging learning on poorly fit datapoints (GT!=prediction)."
> > > > >
> > > > > (The above is my understanding of MaxSup, please do correct me if you feel like it is off.)

---

> > > > > > ### Author Response · Authors · 2024-11-28
> > > > > > **Reply to the additional suggestion from Reviewer 2YeD**
> > > > > >
> > > > > > Thanks for your invaluable suggestion, which significantly enhances the intuitive explanation of MaxSup and we have incorporated your feedback in line 204-209 and line 259-266 in our revised version.

---

> > > > > > > ### Comment · Reviewer_2YeD · 2024-11-29
> > > > > > >
> > > > > > > Thanks for the update, the new (blue) text helps a lot.
> > > > > > >
> > > > > > > My only remaining issue lies with the logit penalty/transfer learning results.
> > > > > > >
> > > > > > > > significant reduction of ImageNet classification accuracy, probably due to incompatibility with the large learning rate and batch size
> > > > > > >
> > > > > > > Kornblith et al. (2021) use a learning rate of 1.6 and a batch size of 4096 (4096/256 * 0.1) and their logit penalty scaling factor is very small (8e-4). With these settings they are able to achieve improved ImageNet accuracy. Did you perhaps set the loss mixing scalar too high?
> > > > > > >
> > > > > > > > Logit Penalty imposes much stronger constraints on the logits.
> > > > > > >
> > > > > > > There's a bit more to it than that I think. A quick bit of napkin math seems to give the per-logit gradient of the logit penalty as $\beta\frac{z_i}{||\boldsymbol{z}||}$, which will be dominated by the max logit, especially when other logits are close to zero (which I personally have observed happens with logits on ImageNet/ResNet at least). In fact, in the case where $z_{max} > 0$ and all other $z = 0$ the logit penality and MaxSup end up with almost exactly the same gradient. Alternatively, you arrive at the same result if you increase the $p$ of the $p$-norm from $2\rightarrow\infty$.
> > > > > > >
> > > > > > > If we consider the results in Kornblith et al. (2021) they show that logit penalty gives significantly worse transfer than even label smoothing. To me this suggests the interesting result that regularising the logits that are not the max logit has a strong negative effect on transfer, but regularising the max logit by itself doesn't.

---

> > > > > > > > ### Comment · Reviewer_2YeD · 2024-11-29
> > > > > > > >
> > > > > > > > On second thought, this could be a promising direction for future work. I think following the above thread about regularising only the max logit vs other logits in the context of transfer learning could potentially better explain the results of Kornblith et al. (2021).

---

> > > > > > > > ### Author Response · Authors · 2024-12-02
> > > > > > > > **Reply to the follow-up discussion with Reviewer 2YeD**
> > > > > > > >
> > > > > > > > We sincerely thank the reviewer for the time and effort during the rebuttal phase. We are pleased to address the remaining issue and suggestion as outlined below:
> > > > > > > >
> > > > > > > > **Regarding ImageNet Accuracy**: Thanks to the Reviewer's suggestion, we have examined that our scaling factor was 4e-3. We have run additional experiments with the scaling factor 8e-4, and the results are as follows:
> > > > > > > > |d_within train | d_all train | R^2 train | d_within val|d_all val |R^2 val |
> > > > > > > > | ---------- | ------------------------- | --------------- | --------------- | ---------- | ------------------------- |
> > > > > > > > |0.2604 | 0.6830 |0.6187 | 0.2853 | 0.6712 | 0.5749|
> > > > > > > >
> > > > > > > > | ImageNet Acc.    | CIFAR10 Transfer Acc. |
> > > > > > > > |------------------|-----------------------|
> > > > > > > > | 76.59  | 0.7209              |
> > > > > > > >
> > > > > > > >
> > > > > > > > **Regarding the insightful feedback about the relation between logit penalty and MaxSup**: Based on the Reviewer's feedback, it might be interesting to analyze the differences between the logit values of the classifiers trained with logit penalty and Maxsup. We are glad to provide such an analysis and incorporate the feedback in the camera ready version.
> > > > > > > >
> > > > > > > > PS: we promise to provide the conditioning number analysis in the camera ready version as well.

---

> > > > > > > > > ### Comment · Reviewer_2YeD · 2024-12-03
> > > > > > > > >
> > > > > > > > > Thanks for the final update. I have accordingly revised my score from 5 to 8. Although this discussion period has been intensive, I hope the authors have come away from it feeling like it was a positive experience and that their paper has been improved as a result.
> > > > > > > > >
> > > > > > > > > One final piece of feedback: I would further emphasise the advantages of the simplicity of MaxSup over existing competing approaches (for example in the introduction and around line 400), as I believe this is a major strength of MaxSup over competing approaches (which you have already demonstrated to be less consistent/more fragile).

---

> > > > > ### Author Response · Authors · 2024-11-28
> > > > > **Reply to follow-up queries from Reviewer 2YeD**
> > > > >
> > > > > We sincerely thank the reviewer for acknowledging the significance of our contribution and for the constructive suggestions aimed at improving the presentation and analysis of our work. We apologize for not adequately addressing the mentioned aspects in the previous revision due to time limit. In response to the reviewer’s valuable feedback, we have carefully revised the manuscript and have addressed all the suggested points in detail, as outlined below:
> > > > >
> > > > > **A1**: We have added the results of L2 Logit Penalty in Table 2 and Table 4. We have also included a discussion about the relation between MaxSup and Logit Penalty in line 318-323. Briefly speaking, Logit Penalty imposes much stronger constraints on the logits, since it reduces the absolute magnitudes of individual logits, while MaxSup only encourages the peak logit value to be close to the mean value of all logits. Similar to the observation in [6], it leads to large class separation but worse linear transfer. In our experiments, it results in a significant reduction of ImageNet classification accuracy, probably due to incompatibility with the large learning rate and batch size, which we adopt for accelerating the training.
> > > > >
> > > > > **A2**: We have now included a clarification in our revised manuscript line 428-429: "In the fine-tuning stage, the vanilla Cross-Entropy loss is used for all models".
> > > > >
> > > > > **A3**: Thanks to the reviewer's suggested literature, we found two kinds of analysis which might help: we are going to perform conditioning number analysis (following [8]) in the coming days to assess whether accelerated convergence can be observed with MaxSup, although the improved feature representation might not necessarily result from improved convergence speed. Besides, recent studies [12,13] show that l2 regularziation imposes low-rank bias on the final layer features and weights of deep unconstrained feature model, which favors solutions with lower-rank than those typically associated with neural collapse. We believe the impact of MaxSup on the optimal solutions is worth exploring for future work.
> > > > >
> > > > > **A4**: We have adopted the same alpha scheduler for both LS and MaxSup by default in our comparisons. There was indeed a typographical error in Table 5 - the correct accuracy should be $76.16\%$ for Label Smoothing without Cutmix and Mixup. Thank you for bringing this to our attention, and we have clarified this point in line 361-363 in the revised version.
> > > > >
> > > > > **A5**: We have improved the clarity of the captions in Fig. 2 by reducing the fontsize and increasing the horizontal space between the subfigures.
> > > > >
> > > > > **A6**: We have added a clarification in line 65-67 in the introduction section: "In line with Zhu et al. (2022), the term ”overconfidence” in our work specifically refers to the network’s confidence in its top-1 prediction, which is different from the overconfidence in the context of model calibration".
> > > > >
> > > > > **A7**: Sorry for the lack of clarity regarding the training setup, we have added a clarification in line 350-353 in the Experiment Setup section.
> > > > >
> > > > > [12] Súkeník, Peter, Marco Mondelli, and Christoph Lampert. "Neural Collapse versus Low-rank Bias: Is Deep Neural Collapse Really Optimal?." NeurIPS 2024
> > > > >
> > > > > [13] Garrod, Connall, and Jonathan P. Keating. "The Persistence of Neural Collapse Despite Low-Rank Bias: An Analytic Perspective Through Unconstrained Features." arXiv preprint arXiv:2410.23169 (2024).

---

> ### Author Response · Authors · 2024-12-04
>
> Dear Reviewer 2YeD,
>
> Thank you once again for your insightful feedback throughout the review process. Your input has significantly contributed to improving the quality of our paper, and we are very pleased with the results.
>
> Best regards,
>
> Authors

---

### Official Review · Reviewer_rCT5 · 2024-11-06

**Soundness:** 2
**Presentation:** 3
**Contribution:** 2
**Rating:** 5
**Confidence:** 4

**Summary:**

The paper proposes a label regularization method called Max Suppression. It shows that vanilla label smoothing has two components: regularization and error-enhancement term. The error enhancement term has negative effects, and the paper proposes an alternate formulation that mitigates its effect by replacing z_gt with z_max.

**Strengths:**

1. The paper is easy to read and well-written.
2. Evaluation is performed on multiple modalities to show its application.
3. An in-depth analysis of the problem of label smoothing was done.

**Weaknesses:**

1. The paper fails to explain the reason behind why replacing z_gt with z_max is the optimal choice.
2. The targets created by the approach are not a probability, and hence, using it with cross-entropy loss may not be correct.
3. Figure 2 is very confusing and difficult to understand.
4. The paper compares the approach across different modalities but should also include results with multiple datasets from the same modality and various architectures (Like done in OLS [2] and Zipf [3])). A lot of the tables could be easily merged to make space for more results.
5. Label Smoothing is known to affect semantic class relationships [1]. The baseline approaches (OLS [2], Zipf [3]) also formulate their loss function to mitigate this, but it is not handled in the proposed approach.
6. In Tables 4 and 5, comparison is only done with the vanilla label smoothing. The improvement is minimal. It would be interesting to see the results of baseline approaches on them.

**Questions:**

1. In the ablation (Table 1), The paper showed removing the Error-Enhancement is beneficial. Why is that not compared/proposed as inplace for replacement?
2. Why is the error-enhancement term having a huge negative impact? The logits of the correct class are generally higher while training.
3. How does it impact the inter-class relationship?

---

> ### Author Response · Authors · 2024-11-23
> **Reply to Reviewer rCT5**
>
> We thank the reviewer for the constructive feedback and address the mentioned points in the following:
>
> > **W1**:  *Why replacing $z_\text{gt}$ with $z_\text{max}$ is the optimal choice?*
>
> This can be explained by our decomposition of the Label Smoothing Loss in equation 6. Thanks to the reviewer's feedback, we have revised Section 3.1 of our manuscript to provide an extended elaboration on this aspect to improve the clarity. We would like to emphasize that **the goal of regularization is to penalize the network for being over-confident about its prediction, i.e., the top-1 position** $z_{max}$. However, Label Smoothing not only introduces an undesired error-enhancement term, but also fails to penalize the network's incorrect prediction, i.e., $z_{max} \neq z_{gt}$ in this case. Therefore, simply dropping the error-enhancement term only avoids enhancing the over-confidence in the incorrect prediction, but fails to regularize it. In contrast, replacing $z_{gt}$ with $z_{max}$ could solve both issues perfectly.
>
> This is also supported by the superior performance of MaxSup: Using the regularization term alone ($75.98\%$) only brings marginal improvement ($+0.07\%$) over Label Smoothing ($75.91\%$), from Table 1. In contrast, MaxSup ($76.12\%$) leads to larger improvement ($0.21\%$) over Label Smoothing ($75.91\%$), from Table 6 (Table 8 in the revised version). In order to improve the clarity, we have added the result of MaxSup in Table 1 in our revised manuscript.
>
> > **W2**: *The targets created by the approach are not a probability, and hence, using it with cross-entropy loss may not be correct.*
>
> Although the targets created by our approach are not a probability, they are **guaranteed to be compatible with cross-entropy based on the decomposed loss component in equation 7**: MaxSup is essentially a **stand-alone regularization loss** in addition to the vanilla cross-entropy loss, and its equivalent label-level formulation in equation 8 is merely intended to show its connection to Label Smoothing. Similarly, Negative Label Smoothing [8] also adopts non-probability targets with cross-entropy as a penalization mechanism, which has been demonstrated to be effective.
>
> > **W3**: *Figure 2 is very confusing and difficult to understand.*
>
> Due to the lack of clarity in the qualitative evaluation in Figure 2, we rely on **quantitative evaluation to demonstrate the improved feature representation with MaxSup** in our revised manuscript, using **intra-class variation** measure $d_\text{within}$ (the larger, the better transfer learning) and **inter-class separability** measure $R^2 = 1 - \frac{d_\text{within}}{d_\text{total}}$ (the larger, the better classification) metrics from [6] (suggested by **Reviewer 2YeD**), respectively:
>
> | Methods    | $d_\text{within} \uparrow$ | $d_\text{total}$ | $R^2 \uparrow$ |
> |------------|----------------------------------|-------------------------|----------------|
> | Baseline   | 0.3114                           | 0.5212                 | 0.4025         |
> | LS         | 0.2632                           | 0.4862                 | 0.4690         |
> | MaxSup     | 0.2926 (**+0.03**)            | 0.6081 (**+0.12**)  | 0.5188 (**+0.05**) |
>
> **Caption**: Quantitative measures for inter-class separability and intra-class variation of feature representations, using ResNet-50 trained on ImageNet-1K. The results are calculated on the ImageNet training set. As analyzed in [6], all regularization losses reduce the relative intra-class variance in the penultimate layer representation space, but our MaxSup suffers from the least reduction of intra-class variance, significantly **improving the intra-class variance as well as the inter-class separability over Label Smoothing**.

---

> ### Author Response · Authors · 2024-11-23
> **Reply to Reviewer rCT5**
>
> > **W4**: *The paper compares the approach across different modalities but should also include results with multiple datasets from the same modality and various architectures (Like done in OLS [2] and Zipf [3]).*
>
>
> We have revised the manuscript with extended evaluation of our approach using various architectures across multiple datasets. **MaxSup consistently achieves the highest accuracy among label smoothing alternatives, whereas OLS and Zipf-LS fail to deliver stable performance**, indicating that the previous empirical justification of such  methods is limited to certain training schemes.
>
>
> | Method                | Resnet-18       | Resnet-50       | Resnet-101      | MobileNetV2     |
> |-----------------------|-----------------|-----------------|-----------------|-----------------|
> | Baseline             | 69.11±0.12      | 76.44±0.10      | 76.00±0.18      | _71.42±0.12_    |
> | Label Smoothing      | 69.38±0.19      | 76.65±0.11      | 77.01±0.15      | 71.40±0.09      |
> | ${\text{Zipf-LS}}$| 69.43±0.13      | _76.89±0.17_    | 76.91±0.14      | 71.24±0.16      |
> | ${\text{OLS}}$   | _69.45±0.15_    | 76.81±0.21      | _77.12±0.17_    | 71.29±0.11      |
> | **MaxSup**           | **69.59±0.13**  | **77.08±0.07**  | **77.33±0.12**  | **71.59±0.17**  |
>
> **Caption**: Performance comparison of classic convolutional neural networks on ImageNet-1K. The training script used was consistent with *TorchVision V1 Scripts*. Note that a larger batch size was employed to accelerate the experimental process, and the learning rate was adjusted based on the linear scaling principle.
>
>
> | Method                | Resnet-18       | Resnet-50       | Resnet-101      | MobileNetV2     |
> |-----------------------|-----------------|-----------------|-----------------|-----------------|
> | Baseline             | 76.16±0.18      | 78.69±0.16      | 79.11±0.21      | 68.06±0.06      |
> | Label Smoothing      | 77.05±0.17      | _78.88±0.13_    | 79.19±0.25      | _69.65±0.08_    |
> | ${\text{Zipf-LS}}$| 76.21±0.12      | 78.75±0.21      | 79.15±0.18      | 69.39±0.08      |
> | ${\text{OLS}}$   | _77.33±0.15_    | 78.79±0.12      | _79.25±0.15_    | 68.91±0.11      |
> | **MaxSup**           | **77.82±0.15**  | **79.15±0.13**  | **79.41±0.19**  | **69.88±0.07**  |
>
> **Caption**: Performance comparison of classic convolutional neural networks on CIFAR100. The training script used was consistent with *TorchVision V1 Scripts*.
>
>
>
> > **W5**: *Label Smoothing is known to affect semantic class relationships [1]. The baseline approaches (OLS [2], Zipf [3]) also formulate their loss function to mitigate this, but it is not handled in the proposed approach.*
>
> MaxSup indeed **improves the feature representation w.r.t. class relationships in a more elegant way**, which can be proved by the improved intra-class variance and inter-class separability with MaxSup (Please refer to our answer to **W3** and **Q3**). In contrast, both OLS and Zipf merely demonstrate improved classification performance over label smoothing, but it is not justified whether they have mitigated the issue of Label Smoothing's semantic class relationships.
>
> Moreover, our work provides **theoretical justification and thoroughly derives a solution to the inherent issue of LS resulting in more confident errors**, which is empirically identified very recently [7]. Our consistently superior performance compared to OLS and Zipf (please refer to our answer to **W4**) also suggests that the potential of such empirical approaches could be inherently limited.
>
>
>
> > **W6**: In Tables 4 and 5, comparison is only done with the vanilla label smoothing. The improvement is minimal. It would be interesting to see the results of baseline approaches on them.
>
> The BLEU score of the baseline approach in Table 4 is 34.3 ± 0.09, so the improvement of Label Smoothing itself is minor, probably due to the limited dataset size. And the **$200$% relative improvement** ($+0.3\%$ with MaxSup vs. $+0.1\%$ with LS) of MaxSup over Label Smoothing should be satisfactory.
>
> The mIoU of the baseline approach is 43.4 in Table 5, so the improvement using Label Smoothing and MaxSup is $0.3$ and $0.7$, respectively. The **over $100$% relative improvement** of MaxSup over Label Smoothing is significant.
>
> To better highlight our improvement, we have added the baseline results in Table 4 and 5, and emphasized the  improvement of our MaxSup over the baseline.
>
>
> > **Q1**: *In the ablation (Table 1), The paper showed removing the Error-Enhancement is beneficial. Why is that not compared/proposed as inplace for replacement?*
>
> When simply removing the Error-Enhancement term, the $+0.07\%$ improvement (baseline + Label Smoothing $75.91\%$ vs. baseline + regularization $75.98\%$) is **marginal**. We provide a detailed explanation on why replacing $z_{gt}$ with $z_{max}$ is optimal in our answer to **W1**.

---

> ### Author Response · Authors · 2024-11-23
> **Reply to Reviewer rCT5**
>
> > **Q2**: *Why is the error-enhancement term having a huge negative impact? The logits of the correct class are generally higher while training.*
>
> The $0.58\%$ accuracy drop (baseline $74.21\%$ vs. baseline + error enhancement $73.63\%$) is reasonable, because the error-enhancement **takes effect since the training started** and the accuracy gradually increases from near 0 to the final accuracy of $73.63\%$. Even when the training converges, there are still around $25$% incorrect predictions.
>
>
> > **Q3**: *How does it impact the inter-class relationship?*
>
> As analyzed in our answer to **W1**, Label Smoothing only performs the desired regularization on the correct predictions (top-1 probability), whereas MaxSup regularizes both the correct and incorrect predictions (top-1 probability), thereby leading to even larger inter-class separability. Please refer to our answer to **W3** for more details. Moreover, MaxSup eliminates the error-enhancement defect of Label Smoothing, which may be the cause of the severely reduced intra-class variance.
>
> **Reference**:
>
> - [6] Kornblith et al. "Why Do Better Loss Functions Lead to Less Transferable Features?" *NeurIPS 2021.
> - [7] Zhu et al. "Rethinking Confidence Calibration for Failure Prediction." *ECCV 2022.*
> - [8] Wei, Jiaheng, et al. "To smooth or not? When label smoothing meets noisy labels." *arXiv preprint arXiv:2106.04149* (2021).

---

> ### Comment · Reviewer_rCT5 · 2024-11-28
> **Follow-up queries**
>
> Thank you for the rebuttal. I was on a family emergency and could not get back in time. I have a few open questions after going through the rebuttal:
>
> W1: Why replacing $z_\text{gt}$ with $z_\text{max}$ is the optimal choice?
> Rv: I see the benefits of using $z_\text{max}$ on performance and understand the intuition behind why the method chose $z_\text{max}$ but having a theoretical reasoning would back it up better. I think a better replacement could be explored or reasoning must be provided as to why is the optimal choice.
>
> W2: The targets created by the approach are not a probability, and hence, using it with cross-entropy loss may not be correct.
> Rv: "Guaranteed to be compatible" is not the right approach. It also needs to make sense mathematically, which states that targets must be probability vectors.
>
> W3: Figure 2 is very confusing and difficult to understand.
> Rv: Thank you for the analysis. It was helpful.
>
> W4: The paper compares the approach across different modalities but should also include results with multiple datasets from the same modality and various architectures (Like done in OLS [2] and Zipf [3]).
> Rv: These results help a lot.
>
> W5: Label Smoothing is known to affect semantic class relationships [1]. The baseline approaches (OLS [2], Zipf [3]) also formulate their loss function to mitigate this, but it is not handled in the proposed approach.
> Rv: The explanation still does not talk about/clarify the impact on class relationships.
>
> W6: In Tables 4 and 5, comparison is only done with the vanilla label smoothing. The improvement is minimal. It would be interesting to see the results of baseline approaches on them.
> Rv: I agree it is an improvement over vanilla baselines but that does not mean it will be better than other alternatives. The baselines must be compared as well.
>
> Q1: In the ablation (Table 1), The paper showed removing the Error-Enhancement is beneficial. Why is that not compared/proposed as inplace for replacement?
> Rv: I understand the improvements. But we need to compare the results on a common setup. Either the results without Error-Enhancement should be included for the results datasets (preferred way) or performance in this setup with the proposed approach.
>
> Q2: Why is the error-enhancement term having a huge negative impact? The logits of the correct class are generally higher while training.
> Rv: Thank you for the explanation.
>
> Q3: How does it impact the inter-class relationship?
> Rv: The explanation still does not talk about/clarify the impact on class relationships.

---

> ### Author Response · Authors · 2024-12-02
> **Reply to the follow-up queries of Reviewer rCT5**
>
> We sincerely thank the reviewer for the follow-up suggestions and questions. We address the raised points in the following:
>
> > **W1**: Why replacing $z_{\text{gt}}$ with $z_{\text{max}}$ is the optimal choice? Rv: I see the benefits of using $z_{\text{max}}$ on performance and understand the intuition behind why the method chose but having a theoretical reasoning would back it up better. I think a better replacement could be explored or reasoning must be provided as to why is the optimal choice.
>
> **A1**: We sincerely appreciate the Reviewer for **acknowledging** the **benefits of MaxSup**, as well as **the rationale behind it**.
>
> As for the term "optimal", we totally **agree with the Reviewer** that reasoning must be provided for such a strong wording. Note that we **never claimed optimality in the paper**, and we apologize for any imprecise uses of the word "optimal" in our answer to Reviewers during the rebuttal. **As pointed out by Reviewer 2YeD in the reply to Reviewer 59zm, it is difficult in and of itself to make any meaningful claims about optimality (in the sense of maximizing some criterion) when we are measuring performance according to generalisation on unseen data**. And our contributions are **independent of the assumption of optimality**. Therefore, "optimality" **shall be less relevant to** the evaluation of our paper.
>
> However, we are pleased to provide more detailed reasoning for proposing MaxSup. Note that MaxSup **only modifies the max and gt positions** of Label Smoothing (Please refer to Eq. 1 and Eq. 8 for comparison), which are **essential** to solve the error-enhancement and inconsistent regularization issues of Label Smoothing: when the network makes incorrect predictions, the gradient of MaxSup loss w.r.t. $z_{gt}$ is $\frac{\partial L_\text{MaxSup}}{\partial z_{gt}}= -\frac{1}{K}\alpha$, whereas the gradient w.r.t. $z_{max}$ is $\frac{\partial L_\text{MaxSup}}{\partial z_{max}}=\frac{K-1}{K}\alpha$(please refer to our Appendix C.2 for more details). The **negative gradient** $-\frac{1}{K}\alpha$ of MaxSup w.r.t. $z_{gt}$  encourages to **increase the underestimated $z_{gt}$**, and the significantly larger gradient $\frac{K-1}{K}\alpha$ of MaxSup w.r.t. $z_{max}$ **correctly focuses on suppressing the overly large $z_{max}$**.
>
> Additional changes are avoided (a small amount of probability mass is still uniformly distributed to the other positions as done in Label Smoothing), since they are irrelevant to the studied issues and their impact is unknown (exploration of alternatives to the uniform label component $\frac{\alpha}{K}$ in Eq. 8 is **orthogonal** to our study). In line with the widely accepted heuristic of **Occam's Razor**, i.e., **Entities should not be multiplied unnecessarily**, we assert that MaxSup is **an elegant solution**.
>
> Please also refer to our follow-up discussion with Reviewer 59zm for the comparison to other alternatives.

---

> ### Author Response · Authors · 2024-12-02
> **Reply to the follow-up queries of Reviewer rCT5**
>
> > **W2**: The targets created by the approach are not a probability, and hence, using it with cross-entropy loss may not be correct. Rv: "Guaranteed to be compatible" is not the right approach. It also needs to make sense mathematically, which states that targets must be probability vectors.
>
> **A2**: We apologize for the lack of clarity in our previous answer, and we are glad to provide a more detailed explanation why MaxSup is **mathematically guaranteed to be compatible with Cross-Entorpy Loss**:
>
> Let $\mathbf{s'}$ denote the soft label created by MaxSup, and $\mathbf{q}$ denote the predicted probability vector, and $\mathbf{y}$ denote the one-hot ground-truth label. It can be easily proven (similar to our proof of Lemma 3.2) that the Cross-Entropy Loss between $\mathbf{s'}$ and $\mathbf{q}$ can be decomposed as follows:
> $$
> L =H(\mathbf{s'}, \mathbf{q}) = H(\mathbf{y}, \mathbf{q}) + L_{\textit{MaxSup}}
> $$
> where the Max Suppression Loss $L_{\textit{MaxSup}}$ is given by:
> $$
> L_{\textit{MaxSup}} = \alpha ( H( \frac{\mathbf{1}}{K}, \mathbf{q} ) - H(1_{k=Argmax(\mathbf{q})}, \mathbf{q})),
> $$
> (sorry that we had problem to show \mathbb correctly at openreview, thus we omit it in our equation)
>
> It can be seen that **training with the soft targets** created by MaxSup is **mathematically equivalent** to **training with a weighted combination of three individual objectives**, i.e., minimizing the Cross-Entropy with one-hot Label (**vanilla loss**) $\mathbf{y}$, where $y_k = 1_{k=\textit{gt}}$ ($y_k = 1$ if class $k$ is the ground-truth class; otherwise $y_k = 0$), minimizing the Cross-Entropy with uniform Label (**the effective component of Label Smoothing loss is kept**), and maximizing the Cross-Entropy with one-hot Label (**the problematic component of Label Smoothing loss is replaced**) $1_{k=\textit{Argmax}(\mathbf{q})}$ ($y_k = 1$ if class $k$ is top-1 prediction; otherwise $y_k = 0$). It is noteworthy that **the minus and plus signs only affect the direction of the optimization, i.e., minimization or maximization**, thus **the individual Cross-Entropies and their sum are always valid**.
>
> In addition, our paper shows that Max Suppression Loss $L_{\textit{MaxSup}}$ is mathematically equivalent to the logit-level form in Eq. 7, and the full training objective can be written as:
> $$
> L = H(\mathbf{y}, \mathbf{q})+  \alpha(z_{\textit{max}}  - \frac{1}{K} \sum^{K}_{k = 1} z_k),
> $$
> it can be also observed from the above equation that the compatibility issue doesn't arise in our approach, since the validity of the vanilla Cross-Entropy is not affected.

---

> ### Author Response · Authors · 2024-12-02
> **Reply to the follow-up queries of Reviewer rCT5**
>
> > **W5**: Label Smoothing is known to affect semantic class relationships [1]. The baseline approaches (OLS [2], Zipf [3]) also formulate their loss function to mitigate this, but it is not handled in the proposed approach. Rv: The explanation still does not talk about/clarify the impact on class relationships.
>
> **A5**: According to [1] (Section 2) and [6] (Section 3), **the semantic class relationships** are exactly characterized by **inter-class separation** / **intra-class variation** of **the learned representations in the penultimate layer**. Although the referred literature [1] is not explicitly given by the reviewer, it should refer to the paper "When does label smoothing help?" by Müller, Rafael, Simon Kornblith, and Geoffrey E. Hinton, because it revealed that "label smoothing encourages the representations of training examples **from the same class to group in tight clusters**. This results in **loss of information** in the logits about **resemblances between instances of different classes**".
>
> If the reviewer did refer to the above issue, then it is both qualitatively and quantitatively validated that MaxSup **alleviates such an issue**, while OLS and Zipf-LS have **not been demonstrated** to mitigate it. We would like to provide a more explicit explanation:
>
> The issue that "the representations of training examples  **from the same class to group in tight clusters**" essentially means that Label Smoothing leads to **small intra-class variation**, which is qualitatively observed via visualization of the penultimate layer representations in [1]. In our visualization (We have improved the clarity of Figure 2, which is put in the appendix as Figure 3 in our revised version), **MaxSup is shown to result in much larger intra-class variation (more scattered clusters of each class) compared to Label Smoothing, while maintaining larger distances between different classes (large inter-class separation benefits classification)**. Moreover, we have also demonstrated the improvement over Label Smoothing through quantitative evaluation. As suggested by Reviewer 2YeD, **such quantitative metrics are also used for evaluating the learned semantic class relationships, but are more convincing than the illustrative plots in [1]**.
>
> > **W6**: In Tables 4 and 5, comparison is only done with the vanilla label smoothing. The improvement is minimal. It would be interesting to see the results of baseline approaches on them. Rv: I agree it is an improvement over vanilla baselines but that does not mean it will be better than other alternatives. The baselines must be compared as well.
>
> **A6**: We thank the reviewer for the constructive suggestion, and are glad to provide the results of other alternatives for the machine translation and semantic segmentation tasks as suggested. We promise to include the results in the camera-ready version.
>
> **Table 1: Comparison of Label Smoothing and MaxSup on IWSLT 2014 German to English Dataset.**
>
> | Model       | Param. | Method          | BLEU Score             |
> | ----------- | ------ | --------------- | ---------------------- |
> | Transformer | 38 M   | Baseline        | 34.3 ± 0.09            |
> |             |        | Label Smoothing | 34.4 (+0.1) ± 0.07     |
> |             |        | OLS             | 34.5 (+0.2) ± 0.06            |
> |             |        | Zipf-LS         | 34.4 (+0.1) ± 0.07            |
> |             |        | **MaxSup**      | **34.6 (+0.3) ± 0.09** |
>
> **Table 2: Comparison of Label Smoothing and MaxSup on ADE20K Validation Set.**
>
> | Backbone   | Segmentation Architecture | Method          | mIoU (MS)       |
> | ---------- | ------------------------- | --------------- | --------------- |
> | DeiT-Small | UperNet                   | Baseline        | 43.4            |
> |            |                           | Label Smoothing | 43.7 (+0.3)     |
> |            |                           | OLS             | 43.8 (+0.4)     |
> |            |                           | Zipf-LS         | 43.8 (+0.4)     |
> |            |                           | **MaxSup**      | **44.1 (+0.7)** |
>
> For the machine translation task, the improvements of Label Smoothing and all the alternatives are relatively small. As we have analyzed in our paper, this probably stems from the constraints of this particular task, e.g., limited dataset size. Thus the comparison on this particular task might be less informative compared to other tasks in our paper.
>
> For the semantic segmentation task, the improvements of OLS and Zipf-LS over the vanilla baseline are limited compared to the improvement of MaxSup, indicating the benefit of the improved feature representation with MaxSup for downstream tasks.

---

> ### Author Response · Authors · 2024-12-02
> **Reply to the follow-up queries of Reviewer rCT5**
>
> > **Q1**: In the ablation (Table 1), The paper showed removing the Error-Enhancement is beneficial. Why is that not compared/proposed as inplace for replacement? Rv: I understand the improvements. But we need to compare the results on a common setup. Either the results without Error-Enhancement should be included for the results datasets (preferred way) or performance in this setup with the proposed approach.
>
> **A1**: We apologize for the lack of clarity in our previous answer. The comparison between the results without Error-Enhancement ($75.98\%$) and our MaxSup ($76.12\%$) in Table 1 in our revised manuscript is under the same setup, i.e., Deit-Small model without CutMix and Mixup. And the results for training Deit-Small (with CutMix and Mixup)  on ImageNet1k without Error-Enhancement and with MaxSup are 79.96±0.13 and 80.16±0.09, respectively.
>
>
> > **Q3**: How does it impact the inter-class relationship? Rv: The explanation still does not talk about/clarify the impact on class relationships.
>
> **A3**: According to [1] (Section 2) and [6] (Section 3),, **the intra-class variation and inter-class separability of the learned representations** reflect  **the impact on inter-class relationships** w.r.t. **the relationship between individual instances of a class to other classes** and **the relationship between the averaged features of each class**, respectively:
>
> - The inter-class separation, i.e., the distances between averaged class features, is further increased compared to Label Smoothing. As analyzed in [1], this indicates improved classification performance.
>
> - The intra-class variation, i.e., "**the information about the resemblances between instances of different classes**" ([1]), is increased compared to Label Smoothing. According to [1], the heavily reduced intra-class variation, or the "overly tight cluster", "results in loss of information in the logits about resemblances between instances of different classes". Therefore, the increased intra-class variation with MaxSup indicates that **such information are better retained**, which benefits knowledge distillation [1] and transfer learning [6].
>
> By examining the intra-class variation and inter-class separability both qualitatively and quantitatively (please refer to our answer to W3), MaxSup is demonstrated to have **a positive impact on the inter-class relationship**, w.r.t. both **the relationship between individual instances of a class to other classes** and **the relationship between the averaged features of each class**.

---

> ### Comment · Reviewer_rCT5 · 2024-12-03
>
> W1: Why replacing $z_\text{gt}$ with $z_\text{max}$ is the optimal choice? Rv: I see the benefits of using $z_\text{max}$ on performance and understand the intuition behind why the method chose $z_\text{max}$ but having a theoretical reasoning would back it up better. I think a better replacement could be explored or reasoning must be provided as to why is the optimal choice.\
> rCT5: I agree with 59zm. Without any theoretical proof/good logic, the proposed approach feels like a hack.
>
> W2: The targets created by the approach are not a probability, and hence, using it with cross-entropy loss may not be correct. Rv: "Guaranteed to be compatible" is not the right approach. It also needs to make sense mathematically, which states that targets must be probability vectors.\
> rCT5: Thank you for clarifying this.
>
> W5: Label Smoothing is known to affect semantic class relationships [1]. The baseline approaches (OLS [2], Zipf [3]) also formulate their loss function to mitigate this, but it is not handled in the proposed approach. Rv: The explanation still does not talk about/clarify the impact on class relationships.\
> rCT5: Yes, in the paper "When does label smoothing help?", the authors showed how the label smoothing destroys inter-class relationships. OLS discusses this in their paper they use a non-uniform vector to mitigate the negative effects. No such discussion /impact/analysis has been done in the paper.
>
> W6: In Tables 4 and 5, comparison is only done with the vanilla label smoothing. The improvement is minimal. It would be interesting to see the results of baseline approaches on them. Rv: I agree it is an improvement over vanilla baselines but that does not mean it will be better than other alternatives. The baselines must be compared as well.\
> rCT5: Thank you for the results.
>
> Q1: In the ablation (Table 1), The paper showed removing the Error-Enhancement is beneficial. Why is that not compared/proposed as inplace for replacement? Rv: I understand the improvements. But we need to compare the results on a common setup. Either the results without Error-Enhancement should be included for the results datasets (preferred way) or performance in this setup with the proposed approach.\
> rCT5: Thank you for the results. These numbers (without Error-Enhancement and with MaxSup are 79.96±0.13 and 80.16±0.09) are too close which makes me wonder if removing Error-Enhancement is a simpler/better solution as it achieves similar results and has a better reasoning.

---

> > ### Author Response · Authors · 2024-12-04
> >
> > We sincerely appreciate the Reviewer’s constructive suggestions and thoughtful feedback during the rebuttal. The suggested experiments have substantially enhanced the quality of our paper. We would also like to take this opportunity to clarify a few misunderstandings.
> >
> > **W1**: Instead of proving MaxSup as an optimal solution, we have provided theoretical proof (please refer to the gradient analysis in Appendix C) that MaxSup is an **effective and elegant solution** to the studied issues of Label Smoothing. Note that **even Label Smoothing and its alternatives, including Zipf-LS and OLS, are not proven as an optimal solution**, yet Label Smoothing is widely applied in training classifiers.
> >
> > **W5**: We kindly request the Reviewer to refer to **Section 4 of our revised manuscript**. As explained in lines 313–317, **the expanded intra-class variation suggests that MaxSup enables the model to capture richer, more detailed similarity information—reflecting how individual examples relate to different classes. In contrast, Label Smoothing tends to ‘erase’ these finer distinctions, as noted by Mueller et al. (2019)**. Specifically, we demonstrate through the significantly improved intra-class distances presented in Table 2 (Section 4) that **MaxSup effectively alleviates the issue** whereby 'Label Smoothing destroys inter-class relationships.' Additionally, we have included a quantitative evaluation of OLS under the same conditions, which reveals that OLS results in much lower intra-class distances and a significant loss in Linear Transfer Performance. This indicates that, contrary to their claim, **OLS does not adequately address the issues associated with Label Smoothing**.
> >
> > | Method | Train(OLS) | Validation(OLS) | Train(MaxSup) | Validation(MaxSup) |
> > | --- | --- | --- | --- | --- |
> > | $d_{within}$ | 0.2707 | 0.2820 | **0.2926** | **0.2998** |
> >
> >
> > | Method | Linear Transfer Acc |
> > | --- | --- |
> > | Baseline | 0.8143 |
> > | Label Smoothing | 0.7458 |
> > | OLS | 0.7821 |
> > | MaxSup | 0.8102 |
> >
> > **Q1**: We would like to provide more results in different setups. It can be seen that Max Sup has a consistent improvement compared to removing Error-Enhancement term.
> >
> > | Model | w/o Label Smoothing | w/o Error-Enhancement | with MaxSup |
> > | --- | --- | --- | --- |
> > | Deit-Small(w/o CutMix&Mixup) | 74.21 | 75.98 | **76.12** |
> > | Deit-Small(with CutMix&Mixup) | 79.69±0.11 | 79.96±0.13 | **80.16±0.09**|
> > | ResNet50 | 76.44±0.10 | 75.84±0.18 | **77.08±0.07** |
> >
> > We hope our explanation adequately addresses the Reviewer’s remaining concerns.

---

### Author Response · Authors · 2024-11-25
**General Comment**

Dear Reviewers,

We are pleased to submit the revised manuscript for your review. All comments have been addressed, and the necessary updates have been incorporated to enhance the quality of the manuscript.

Thank you for your time and consideration.

Best regards,

The Authors

---

### Author Response · Authors · 2024-11-28
**Updated Manuscript**

Dear Reviewers,

We sincerely appreciate your valuable follow-up discussion and thoughtful feedback. Based on your suggestions, we have made further improvements to the manuscript and look forward to any additional discussion.

Although the revision submission deadline is approaching, we remain committed to continuously enhancing the quality of our work and will gladly incorporate any further feedback in the camera-ready version.

Thank you for your time and consideration.

Best regards,

The Authors

---

### Meta-Review · Area_Chair_8Rni · 2024-12-21

**Metareview:**

Label Smoothing is a common regularization technique used to mitigate overconfidence in predictions, but it has drawbacks, such as overly tight clustering of intra-class features and increased confidence in incorrect predictions. The paper introduces Max Suppression (MaxSup), a proposed alternative to Label Smoothing (LS) for improving feature representation in neural networks.

Strengths:
- This paper aims to address the drawbacks of label smoothing, which is an interesting work.
- The experimental results cover diverse tasks, including image classification, machine translation, and semantic segmentation.
- This paper provides a link to the codebase, which supports the reproducibility of this paper.

Reasons to reject:

- The proposed method seems to be only an engineering refinement. I understand that we should appreciate that a simple trick can work well, but this should be supported by theoretical guarantees and promising experimental performance. Unfortunately, this paper do not have theoretical guarantees and promising experimental performance.

- No theoretical guarantees are provided.

- The experimental performance is not so good. We can see that the improvements over compared methods are marginal.

- The writing of this paper is somewhat unclear, especially for the motivation of the proposed method.

**Additional Comments On Reviewer Discussion:**

This paper finally receives the scores of 8 (Reviewer 2YeD), 5 (Reviewer rCT5), 5 (Reviewer 59zm), 5 (Reviewer FzkC). After the rebuttal process, some of the reviewers' concerns were addressed. However, major concerns still remain unsolved. Among those concerns, I especially agree with Reviewer 59zm that the novelty/contribution of this paper is limited. The proposed method seems to be only an engineering refinement. I understand that we should appreciate that a simple trick can work well, but this should be supported by theoretical guarantees and promising experimental performance. Unfortunately, this paper do not have theoretical guarantees and promising experimental performance.

Considering the above situation and given that nearly all the reviewers vote for weak reject, I have to commend rejecting this paper.

---

### Decision · Program_Chairs · 2025-01-22

Reject